# Deregulated Gab2 phosphorylation mediates aberrant AKT and STAT3 signaling upon *PIK3R1* loss in ovarian cancer

Xinran Li[1], Victor C.Y. Mak[1], Yuan Zhou[1], Chao Wang[2], Esther S.Y. Wong[3], Rakesh Sharma[4], Yiling Lu[5], Annie N.Y. Cheung[3], Gordon B. Mills[6] & Lydia W.T. Cheung[1]

Copy number loss of *PIK3R1* (p85α) most commonly occurs in ovarian cancer among all cancer types. Here we report that ovarian cancer cells manifest a spectrum of tumorigenic phenotypes upon knockdown of *PIK3R1*. *PIK3R1* loss activates AKT and p110-independent JAK2/STAT3 signaling through inducing changes in the phosphorylation of the docking protein Gab2, thereby relieving the negative inhibition on AKT and promoting the assembly of JAK2/STAT3 signalosome, respectively. Additional mechanisms leading to AKT activation include enhanced p110α kinase activity and a decrease in PTEN level. *PIK3R1* loss renders ovarian cancer cells vulnerable to inhibition of AKT or JAK2/STAT3. The combination of AKT and STAT3 inhibitors significantly increases the anti-tumor effect compared to single-agent treatments. Together, our findings provide a rationale for mechanism-based therapeutic approach that targets tumors with loss of *PIK3R1*.

[1] School of Biomedical Sciences, Li Ka Shing Faculty of Medicine, The University of Hong Kong, Hong Kong, Hong Kong. [2] Department of Obstetrics and Gynecology, Obstetrics and Gynecology Hospital, Fudan University, Shanghai, China. [3] Department of Pathology, Li Ka Shing Faculty of Medicine, The University of Hong Kong, Hong Kong, Hong Kong. [4] Proteomics & Metabolomics Core Facility, Li Ka Shing Faculty of Medicine, The University of Hong Kong, Hong Kong, Hong Kong. [5] Department of Systems Biology, University of Texas MD Anderson Cancer Center, Houston, TX 77030, USA. [6] Knight Cancer Institute, Oregon Health & Science University, Portland, OR 97239, USA. Correspondence and requests for materials should be addressed to L.W.T.C. (email: lydiacwt@hku.hk)

Aberrations of members along the PI3K pathway are among the most common oncogenic events in cancers[1]. The majority of these aberrations occur in the upstream components of the pathway, such as PIK3CA (p110α catalytic subunit of PI3K), PIK3R1 (p85α regulatory subunit of PI3K), and PTEN. So far, most characterized aberrations of these genes result in elevated production of phosphatidylinositol 3,4,5-triphosphate (PIP3) that activates AKT, which is also frequently aberrant in cancers. We and others have shown that p85α mutants that fail to inhibit p110α activity are associated with constitutive PI3K pathway activation[2–4]. Apart from engaging in a p110-bound heterodimer that inhibits and stabilizes p110α, p85α negatively regulates the PI3K pathway by forming homodimer that stabilizes PTEN[5,6], indicating an additional layer of PI3K pathway regulation by p85α to the canonical regulation of p110. Intriguingly, PIK3R1 driver mutations that disrupt the homodimerization lead to PTEN instability and AKT activation.

In line with the proposed tumor-suppressive roles of p85α, PIK3R1 copy number loss is often detected in multiple tumor types including cancers of prostate, ovary, lung and breast. PIK3R1 mRNA expression is also significantly decreased in many of these tumor types, compared with the corresponding normal tissues[7,8]. Reduced PIK3R1 expression associates with poorer survival of breast cancer patients and tumorigenic transformation in breast cancer models[7,9]. The reduced p85α levels lead to increase in classical AKT signaling which mediates these tumorigenic phenotypes[10]. Similar observations were reported in hepatocellular carcinoma mouse models with liver-specific PIK3R1 deficiency wherein these mice had an increase in tumor development[8]. However, in the context of prostate tumorigenesis in which androgen signaling pathway is essential, PIK3R1 depletion inhibits AKT phosphorylation and prostate cancer cell proliferation[11]. Emerging evidence has shown that similar to mutations in PIK3R1 or in other PI3K pathway components[12,13], PIK3R1 loss can induce downstream signaling beyond the canonical AKT pathway. In PIK3R1-depleted renal cancer cells, signaling crosstalk initiated by AKT promotes WNT/β-catenin activation and acquisition of cancer stem-like phenotype[14]. Collectively, these findings underscore the lineage-specific consequences of PIK3R1 loss in cancers.

Ovarian cancer has the most frequent PIK3R1 heterozygous and homozygous deletion across all tumor types in The Cancer Genome Atlas (TCGA)[15,16]. Given the high occurrence of copy number loss and the context-dependent molecular manifestations of the aberration in different cancer lineages, we sought to determine the functional role and therapeutic implication of PIK3R1 loss in ovarian cancer. Here we established that PIK3R1 loss favors ovarian tumorigenesis through co-activation of AKT and JAK2/STAT3 signaling. Further, the activated signaling creates a targetable therapeutic vulnerability in PIK3R1 loss-bearing ovarian cancer cells.

## Results

### PIK3R1 loss promotes acquisition of tumorigenic hallmarks.
PIK3R1 copy number loss was the most frequent in serous ovarian cancer across TCGA[15,16]. In total, 3.5% (20/579) and 68.4% (396/579) tumors had homozygous and heterozygous loss, respectively (Supplementary Fig. 1a). PIK3R1 copy number significantly correlated with mRNA levels ($r = 0.3$; $p < 0.05$) (Supplementary Fig. 1b), suggesting that copy number is a causal factor of changes in expression. We then assessed the functional consequence of p85α downregulation in serous ovarian cancer cell lines (SKOV3, OVCAR8, and OAW28), which express high level of p85α and carry a wild-type PIK3R1 gene. The efficiency of the PIK3R1 siRNA was confirmed by western blotting

(Supplementary Fig. 1c). We observed marked increase in cell proliferation induced by two distinct PIK3R1 siRNA sequences consistently in the three cell lines (Fig. 1a). Cell cycle analysis of synchronized SKOV3 cells suggested that the increased cell proliferation is likely linked to accelerated cell cycle progression. PIK3R1 siRNA-transfected cells showed decreased percentage in G0/G1 phase with a concomitant increased percentage in S and G2/M phases (Fig. 1b). PIK3R1 loss also protected SKOV3 cells from serum depletion-induced apoptosis (Fig. 1c). Further, in vitro cell migration and cell invasion were significantly promoted in PIK3R1 siRNA-transfected cells (Fig. 1d, e). It is noteworthy that cell migration and invasion were assayed 24 h after siRNA transfection, at which time changes in proliferation was negligible.

The observed in vitro oncogenic phenotypes prompted us to evaluate the effect of PIK3R1 loss on tumorigenic progression in vivo. SKOV3 cells stably expressing PIK3R1 shRNA, which consistently had higher viability as demonstrated by colony formation assay (Supplementary Fig. 1d), were injected i.p. into female athymic nude mice. Peritoneal dissemination of tumors, which is a characteristic of ovarian cancer, was assessed by number and weight of peritoneal disseminated tumor nodules formed. Significantly, the tumor burden of PIK3R1 shRNA tumors was higher than that of tumors expressing vector control (Fig. 1f), indicating that PIK3R1 downregulation enhances tumorigenesis and metastatic dissemination. Two stable clones of SKOV3 generated by two independent shRNA sequences displayed similar phenotypes. Collectively, these data indicate that ovarian cancer cells with PIK3R1 loss demonstrate multiple tumorigenic properties, providing an explanation of the frequent PIK3R1 copy number loss in the disease.

### STAT3 and AKT signaling are activated upon PIK3R1 loss.
To decipher the downstream signaling caused by p85α down-regulation, seven serous ovarian cancer cell lines transfected with PIK3R1 siRNA were subjected to reverse phase protein array (RPPA) measuring the levels of 302 key cancer-related proteins. PIK3R1 is mutated in one cell line (OVCAR3) and is wild-type in the remaining six cell lines. A heatmap of proteins with ≥20% difference in expression between PIK3R1 siRNA- and mock-transfected cells across at least three PIK3R1-wild-type cell lines is depicted (Fig. 2a). Proteins that were downregulated upon PIK3R1 loss across cell lines include p85 (as expected) as well as p110α and PTEN, which are known to be stabilized and regulated by p110-bound p85α and p110-unbound p85α homodimer, respectively. Phosphorylation of AKT and its downstream target, the ribosomal protein S6, was increased by PIK3R1 siRNA, except in OVCAR3 that contains PIK3R1 mutation that has been demonstrated to activate AKT[17]. This is concordant with previous studies of other cancer types wherein decreased p85α leads to AKT activation[8,10]. It was shown in renal cancer cells that PIK3R1 loss increases protein amounts of β-catenin[14]. We did not detect changes in expression or degradation-mediating phosphorylation (T41/S45) of β-catenin in any PIK3R1-depleted ovarian cancer cell lines, suggesting a tissue specific function of PIK3R1 loss (Supplementary Fig. 2a). However, to our surprise based on the ability of RPPA to probe multiple pathways, we observed increased STAT3 phosphorylation in PIK3R1-wild-type cell lines. Importantly, the RPPA data were confirmed by western blotting (Fig. 2b). The increases in p-STAT3 and p-AKT in PIK3R1-depleted cells were statistically significant in three independent experiments. The activation of AKT and STAT3 was also recapitulated in the stable PIK3R1 shRNA SKOV3 cells (Supplementary Fig. 2b). The protein levels of MAPK pathway members (ERK1/2, p38 MAPK, and JNK) and STAT5 were not

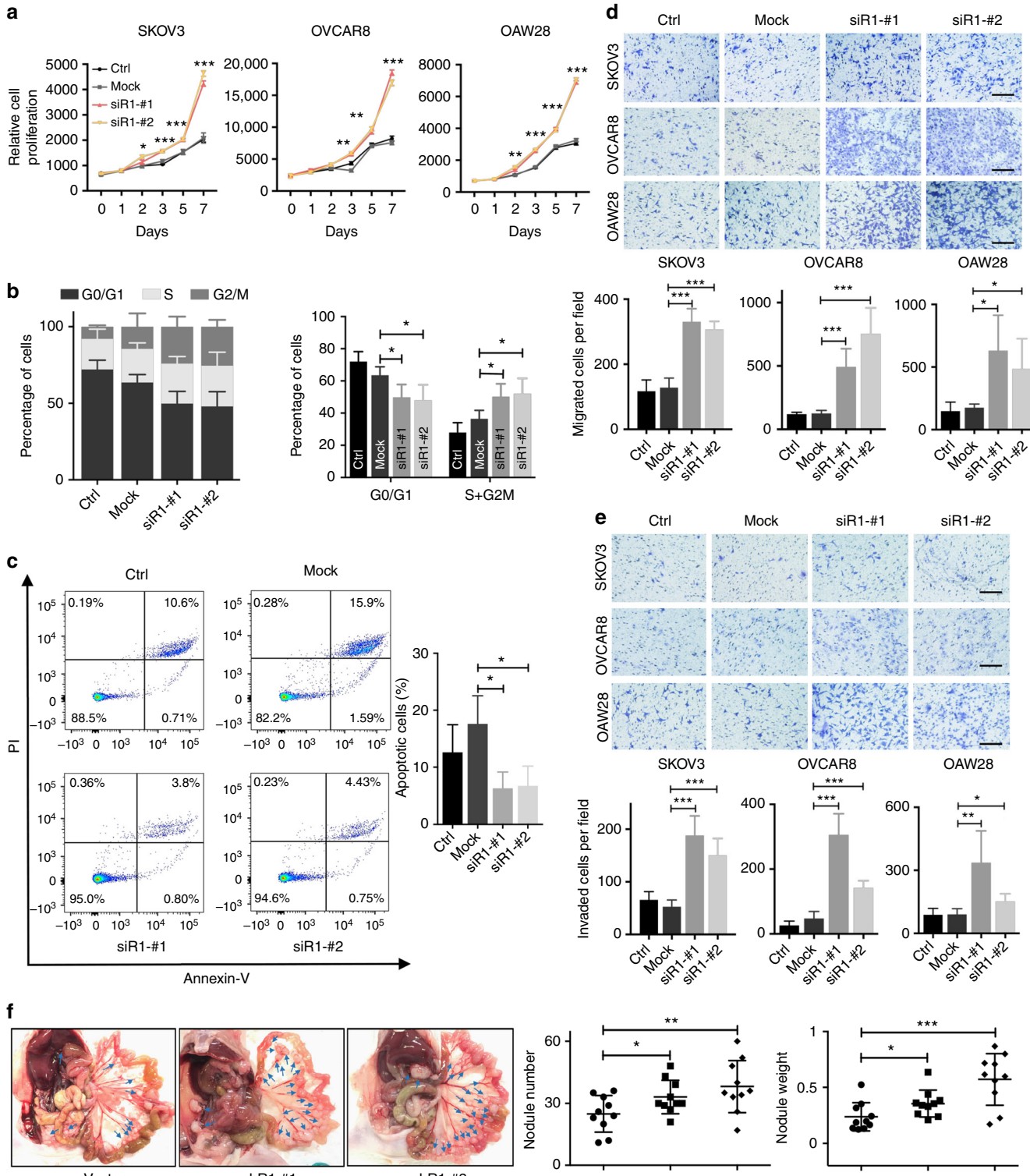

**Fig. 1** *PIK3R1* loss promotes ovarian cancer tumorigenic phenotypes in vitro and in vivo. **a** Ovarian cancer cells (SKOV3, OVCAR8, OAW28) were transfected with siRNA for 24 h before cell seeding. Cell viability was measured over 7 days. **b** Synchronized SKOV3 cells were transfected with siRNA for 48 h before cell cycle analysis. **c** Transfected SKOV3 cells were cultured in FBS-free medium 48 h before apoptosis assay. **d, e** Representative images (upper) and mean numbers of migrated (**d**) or invaded (**e**) ovarian cancer cells (SKOV3, OVCAR8, OAW28) of five fields at magnification of 100 × (lower). Scale bar, 200 μm. **f** SKOV3 cells stably expressing *PIK3R1* shRNA or empty vector were intraperitoneally injected into nude mice (*n* = 10) for 5 weeks. Representative images showing tumor nodules indicated by blue arrows (left), and total nodule weight and number are shown (right). Error bars represent SD. *$p < 0.05$; **$p < 0.005$; ***$p < 0.001$ compared with mock or vector using *t*-test

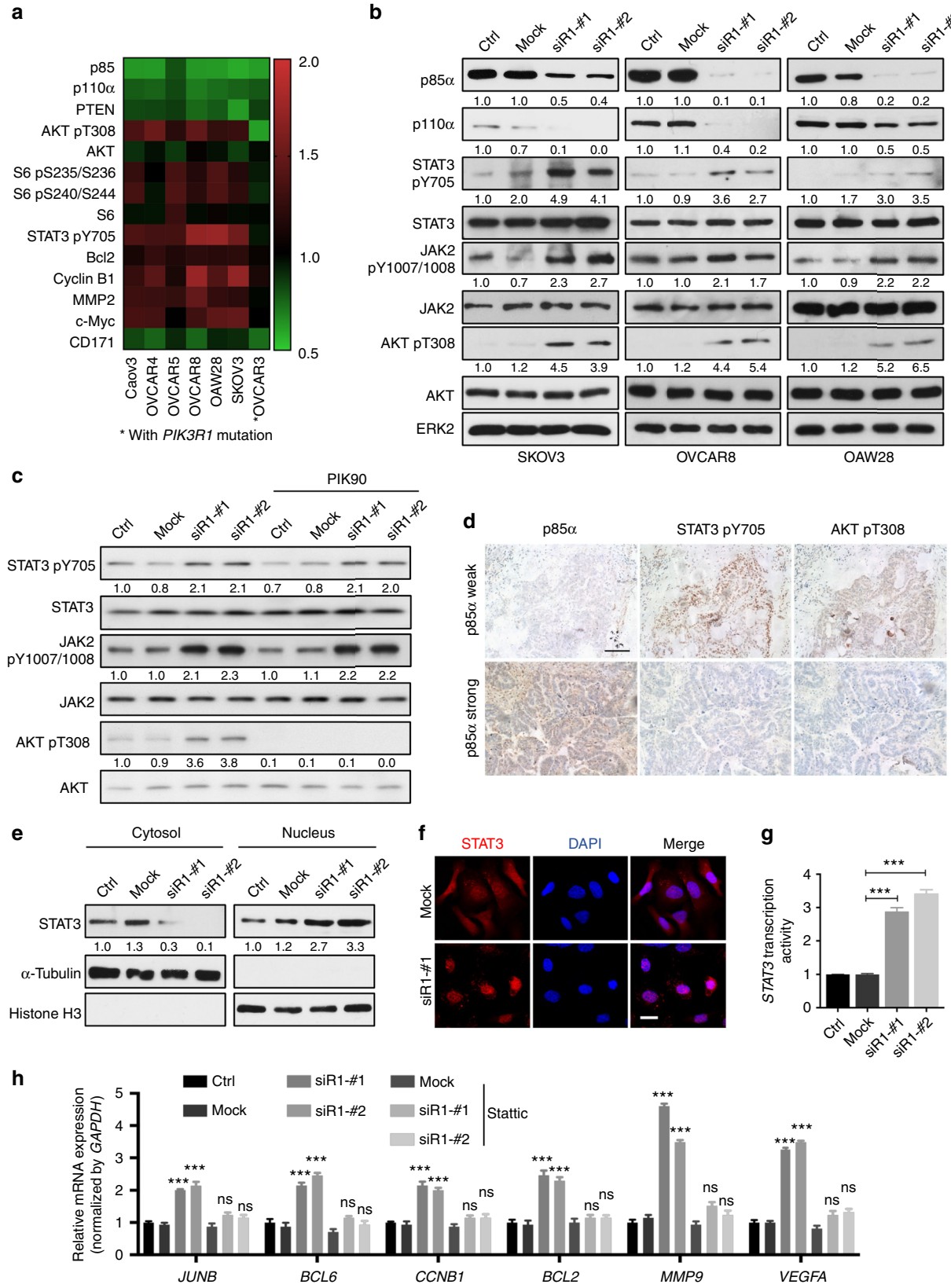

altered in these cells (Supplementary Fig. 2a and Supplementary Fig. 2c).

STAT3 is a substrate for the non-receptor tyrosine kinases JAK2 and Src[18]. An antibody for p-JAK2 was not included in the RPPA. Our western blot data showed that p-JAK2 was increased in *PIK3R1* siRNA-transfected cell lines as well as in stable *PIK3R1* shRNA SKOV3 (Fig. 2b; Supplementary Fig. 2b). In contrast, both RPPA and western blot clearly showed that phosphorylation of Src was not altered upon *PIK3R1* loss (Supplementary Fig. 2a and Supplementary Fig. 2c). These data suggested that *PIK3R1*

**Fig. 2** *PIK3R1* loss activates STAT3 and AKT signaling. **a** Reverse phase protein array was performed using lysates of seven serous ovarian cancer cell lines transfected with *PIK3R1* siRNA for 72 h. Proteins with ≥20% difference in levels between *PIK3R1* siRNA-transfected compared to that of mock-transfected cells in at least three *PIK3R1*-wild-type cell lines are shown. The corresponding total proteins of the phosphorylated proteins are also presented. **b** Total lysates of transfected SKOV3, OVCAR8 or OAW28 were analyzed with western blotting. ERK2 was loading control. **c** SKOV3 cells were transfected with siRNA for 48 h prior to treatment with PIK90 (1 μM) or DMSO for another 24 h. Cell lysates were subjected to western blotting. **d** Representative immunohistochemical images of ovarian cancer specimens stained with p85α, STAT3 pY705 or AKT pT308 antibodies. Nuclei were counterstained in hematoxylin. Scale bar, 100 μm. **e** SKOV3 cells were transfected with siRNA for 72 h and harvested for subcellular fractionation followed by western blotting. α-Tubulin and Histone H3 were molecular markers for the cytosolic and nuclear fractions, respectively. **f** SKOV3 were transfected with siRNA for 24 h and immunofluorescence were performed with STAT3 antibody (Texas red) and DAPI (blue). Scale bar, 20 μm. **g** SKOV3 cells were transfected with *PIK3R1* siRNA 24 h followed by co-transfection of 4xM67 pTATA TK-Luc (containing STAT3-binding site) with Renilla luciferase vector. Dual-luciferase reporter assay was performed after 24 h. **h** SKOV3 transfected with siRNA for 48 h were treated with Stattic (10 μM) or DMSO for 24 h. Total RNA was collected for real-time PCR. Comparative Ct values calculation was performed for the relative abundance of genes with respect to *GAPDH* expression. The numbers below the blots represent the mean values from densitometry readings of three independent experiments. Error bars represent SD. ***$p < 0.001$; ns, no significant difference compared with mock using *t*-test

---

loss induces Src-independent but JAK2-dependent activation of STAT3. Since p85α can function through p110-dependent or -independent mechanisms, we determined the involvement of p110 activity in the induction of STAT3 signaling. A pan PI3K inhibitor (PIK-90), which targets the different p110 isoforms, did not impede the phosphorylation of STAT3 (Fig. 2c). This suggested that *PIK3R1* activates STAT3 signaling in p110-independent manner.

We next examined whether these signaling alterations can also be observed in TCGA ovarian cancer samples. The samples were stratified by *PIK3R1* mRNA levels with those in the lowest and highest quartile defined as *PIK3R1*-low and *PIK3R1*-high, respectively ($n = 103$ in each group). In accord with the cell line data, *PIK3R1*-low patients displayed higher p-STAT3 ($p = 0.058$) and p-AKT ($p = 0.055$) compared to *PIK3R1*-high patients (Supplementary Fig. 2d). Intriguingly, the levels of p-STAT3 and p-AKT significantly correlated in *PIK3R1*-low samples ($r = 0.3$; $p = 0.03$) but not in *PIK3R1*-high samples ($r = 0.08$; $p = 0.5$) (Supplementary Fig. 2e). Expression of p110α and PTEN also tended to be downregulated in *PIK3R1*-low tumors, albeit not statistically significant. We further evaluated the linkage between p85α level and status of AKT and STAT3 activation in an independent set of in-house serous ovarian cancer samples ($n = 43$). Immunohistochemistry staining showed that 31 (72%) tumors had low p85α level. Among these p85α-low tumors, 18 (58%) and 22 (71%) showed strong nuclear p-STAT3 and p-AKT staining, respectively, with 12 (39%) concurrently displayed high levels of p-STAT3 and p-AKT. Representative sections are shown in Fig. 2d. There was a statistically significant negative association between p85α and p-STAT3 ($r = -0.53$, $p = 0.0002$) or p-AKT ($r = -0.4$, $p = 0.007$).

STAT3 is a transcription factor that regulates expression of genes involved in cellular processes such as proliferation, apoptosis, metastasis, and inflammation. We therefore explored the impact of *PIK3R1* loss on STAT3 nuclear trafficking, STAT3 transcriptional activity and STAT3-dependent gene transcription. We observed a pronounced change in the localization of STAT3 upon *PIK3R1* loss using subcellular fractionation. As shown in Fig. 2e, *PIK3R1* siRNA led to translocation of STAT3 to the nucleus from the cytoplasm. This finding was further supported by fluorescence microscopy (Fig. 2f). STAT3 transcriptional activity was measured using a firefly luciferase reporter harboring four copies of STAT3-binding sites. Notably, *PIK3R1* siRNA promoted transcriptional activity of STAT3 by threefold (Fig. 2g). We next assessed mRNA expression of 17 well-established STAT3 transcriptional targets by real-time PCR[19]. Intriguingly, consistent with the increment in cell proliferation and metastasis as shown in Fig. 1, *PIK3R1* siRNA induced expression of genes involved in cell cycle and anti-apoptosis (*CCNB1* and *BCL2*) and

metastasis (*MMP9* and *VEGFA*) (Fig. 2h). Remarkably, cyclin B1 and Bcl2 proteins were also shown to be increased by RPPA (Fig. 2a). Other known targets that were upregulated upon *PIK3R1* loss included transcription factors (*JUNB* and *BCL6*). All these increases were abolished in the presence of a STAT3 inhibitor, indicating that the regulation is indeed mediated through the STAT3 pathway. The mRNA transcripts of *MMP2* and *MYC* (corresponding proteins increased upon *PIK3R1* loss as shown in Fig. 2a) were also increased (Supplementary Fig. 3a), which were partially inhibited by the STAT3 or AKT inhibitor, suggesting that both STAT3 and AKT signaling contributed to the increases. The high concordance between the real-time PCR and RPPA data was also supported by the consistently unchanged levels of cyclin D1, Mcl-1, and Cox-2 (Supplementary Fig. 3b–d). Genes involved in inflammation were not changed (Supplementary Fig. 3c), implying that *PIK3R1* loss unlikely has significant influence on inflammatory response in these cell lines.

**Gab2 mediates downstream signaling of *PIK3R1* loss.** p85α has not been previously shown to impinge on STAT3 signaling and the mechanism of STAT3 activation upon *PIK3R1* loss was unknown. Grb2-associated binder 2 (Gab2) is a scaffold that interacts with signaling molecules including p85α and coordinates signaling events[20,21], prompting us to consider the role of Gab2 in the signaling alterations upon *PIK3R1* loss. Two independent *GAB2* siRNA sequences alone led to decreased AKT phosphorylation. *PIK3R1* loss-induced AKT activation was inhibited by the *GAB2* siRNA (Fig. 3a). Strikingly, *PIK3R1* loss-induced STAT3 and JAK2 phosphorylation was abolished in the presence of the *GAB2* siRNA (Fig. 3a). Further, STAT3 transcriptional activity and STAT3-mediated gene transcriptional program downstream of *PIK3R1* loss were abrogated by *GAB2* siRNA (Fig. 3b, c). These results suggested that Gab2 is involved in the activation of AKT and STAT3 signaling upon *PIK3R1* loss. In contrast, silencing *GAB1*, which is implicated in mediating PI3K and STAT3 activation upon receptor tyrosine kinase (RTK) activation[22,23], did not attenuate *PIK3R1* loss-induced signaling (Supplementary Fig. 4a). Phosphorylation of Gab1 at Y307, that was shown to correlate with STAT3 activation[22], was unaltered upon *PIK3R1* loss (Supplementary Fig. 4b).

The pivotal role of Gab2 in mediating the oncogenic signaling of *PIK3R1* loss is further supported by the observations that *PIK3R1* siRNA failed to promote cell proliferation, cell migration and invasion in the presence of *GAB2* siRNA (Fig. 3d, e; Supplementary Fig. 4c–e). Gab2 binds both STAT3 and JAK2[24,25]. Interestingly, both immunoprecipitation experiments and in situ proximity ligation assay (PLA) showed clearly that *PIK3R1* loss promoted the interaction between Gab2, STAT3, and JAK2 (Fig. 3f, g), suggesting that the enhanced formation of a

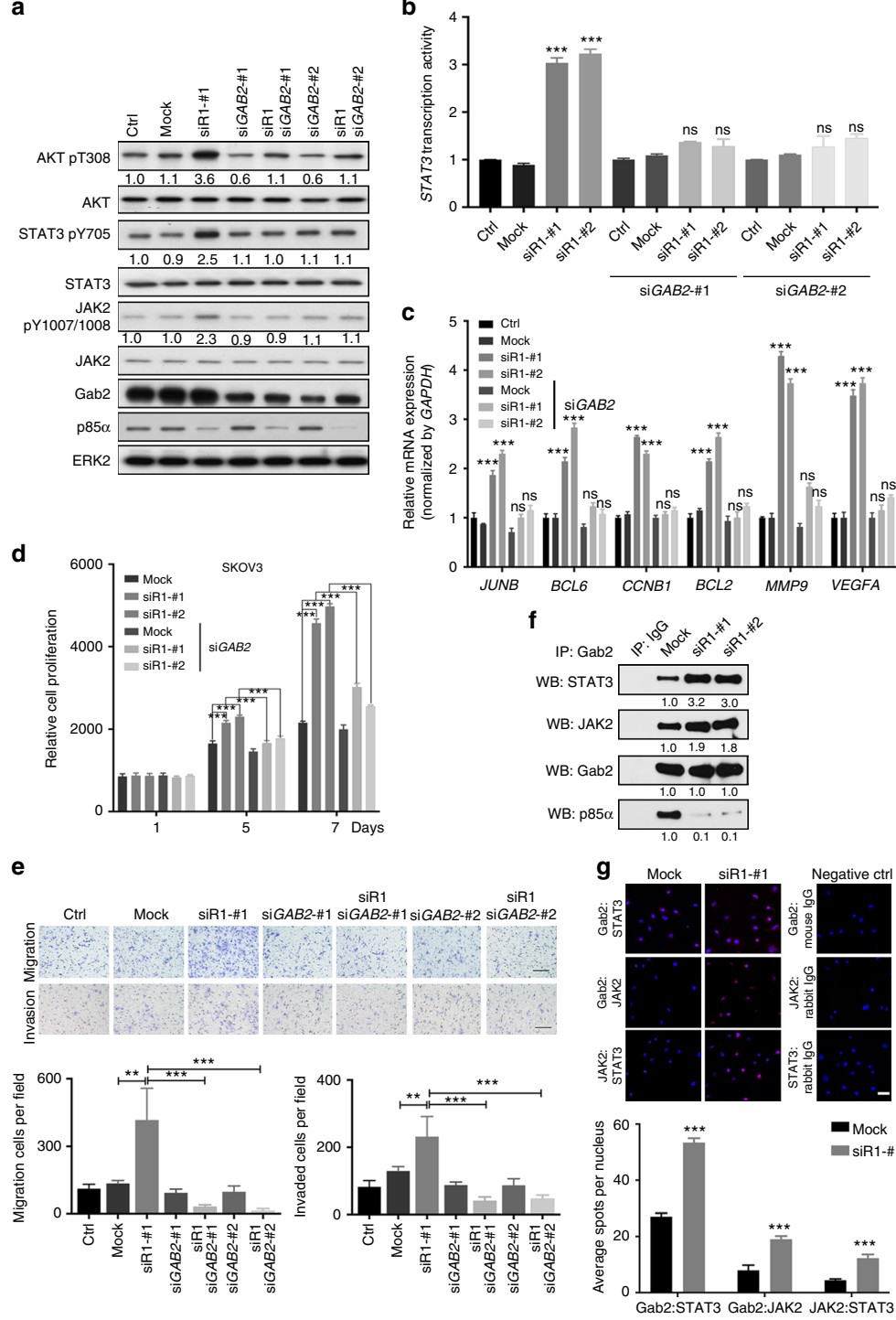

**Fig. 3** Gab2 is required in *PIK3R1* loss-induced signaling and oncogenic phenotypes. **a** Total cell lysate of SKOV3 cells transfected with either *PIK3R1* siRNA, *GAB2* siRNA or in combination for 72 h were harvested for western blotting. **b** SKOV3 cells were transfected with *PIK3R1* siRNA or co-transfected with *GAB2* siRNA. After 24 h, cells were transfected with 4xM67 pTATA TK-Luc (containing STAT3-binding site) with Renilla luciferase vector for another 24 h. Dual-luciferase reporter assay was performed. **c** Total RNA of SKOV3 cell transfected with *PIK3R1* siRNA or in combination with *GAB2* siRNA for 72 h were harvested for real-time PCR. **d** SKOV3 cells were transfected with *PIK3R1* siRNA or co-transfected with *GAB2* siRNA, cell viability was measured over 7 days. **e** Representative images of migrated or invaded SKOV3 cells (upper) and means with SD of migrated or invaded cells of five fields at magnification of 100 × (lower). Scale bar, 200 μm. **f** Total lysates of transfected SKOV3 cells were immunoprecipitated with Gab2 antibody prior to western blotting. **g** SKOV3 cells were transfected with *PIK3R1* siRNA prior to proximity ligation assay with the indicated pairs of antibodies. Representative images are presented (upper) with numbers of averaged spots per nucleus (lower). Scale bar, 50 μm. The numbers below the blots represent the mean values from densitometry readings of three independent experiments. Error bars represent SD. **$p < 0.005$; ***$p < 0.001$; ns, no significant difference compared with mock using *t*-test

JAK2/STAT3 signalosome contributes to activation of the pathway. This increase in signalosome assembly was unlikely due to changes in total Gab2 level because *PIK3R1* siRNA did not affect the expression of Gab2 protein.

**Gab2 S210 and S623 involve in the signaling of *PIK3R1* loss**. The interaction of Gab2 with other signaling molecules depends on Gab2 phosphorylation status. For example, phosphorylation of Gab2 on tyrosine residues Y452, Y476, and Y584 has been implicated in binding to the SH2 domains of p85α[26]. We therefore analyzed the phosphorylation status of Gab2 associated with *PIK3R1* loss. Western blotting was performed using commercially available antibodies of p-Gab2, including S159, Y452, S623, and Y643. First, *PIK3R1* siRNA caused remarkable decrease in Y452 phosphorylation (Fig. 4a), which was rescued by the introduction of p85α SH2 domains or full-length wild-type p85α (Supplementary Fig. 5a), implicating that the decreased phosphorylation could be an effect of reduced association with p85α. Interestingly, we also observed an increase in phosphorylated S623 (Fig. 4a), which was shown to involve in STAT5 activation due to reduced interaction between Gab2 and SHP-2[27]. There was no significant change in the levels of S159 (mediates negative feedback of RTK signaling[28]) and Y643 (involves in ERK1/2 activation[24]).

In an attempt to reveal other phosphorylation sites that are altered, mass spectrometry-based quantitative phosphoproteomics was applied. The analysis confidently yielded two Gab2 phosphopeptides corresponding to two phosphorylation sites (S210 and S405), which were both decreased by *PIK3R1* siRNA (Fig. 4b; Supplementary Fig. 5b). A previously-reported antibody against Gab2 S210[29] was then obtained and its decrease upon *PIK3R1* loss could be validated (Fig. 4c). S210 binds 14-3-3 resulting in inhibition of Gab2-mediated AKT signaling[29]. The role of S405, which was shown to be phosphorylated in IL-2-stimulated T-lymphocytes[30], is unclear at present.

To evaluate the functional significance of these four altered amino acid residues (S210, S405, Y452, and S623) in *PIK3R1* loss-induced signaling, we generated phosphorylation-defective *GAB2* mutants for expression in OVCAR5 cells which have minimal endogenous Gab2 protein. *PIK3R1* siRNA led to statistically significant AKT activation in the parental OVCAR5 without Gab2 expression in three independent experiments (Fig. 4d), suggesting additional Gab2-independent mechanism underlying the activation. This AKT activation is unlikely due to any residual level of Gab2 in the parental OVCAR5 because *GAB2* siRNA could not abolish the increase (Supplementary Fig. 5c). Phosphorylation of AKT was inhibited in cells expressing Y452F mutant regardless of p85α level, confirming a critical role of Y452 in AKT activation (Fig. 4d). The introduction of the S210A mutant, which mimics decreased S210 phosphorylation upon *PIK3R1* loss, led to statistically significant enhanced AKT phosphorylation with or without *PIK3R1* siRNA. In contrast, AKT was activated by *PIK3R1* siRNA to a similar extent in cells with wild-type Gab2, S405A or S623A mutants.

Gab2 expression is crucial for STAT3 and JAK2 activation by *PIK3R1* depletion because these two molecules were not significantly induced by *PIK3R1* siRNA in parental OVCAR5. Strikingly, STAT3 and JAK2 activation was consistently observed upon *PIK3R1* loss after expression of wild-type Gab2, S210A, S405A, or Y452A but not S623A mutant. This highlights the importance of S623 in mediating JAK2/STAT3 activation. In addition, the amounts of JAK2 and STAT3 immunoprecipitated with the Gab2 S623A mutant were lower than those with wild-type Gab2 (Fig. 4e), indicating the ability to assemble JAK2/

STAT3 signalosome was impaired in the Gab2 S623A mutant. Since phosphorylation of S210 and S623 might lead to altered binding to signaling regulatory proteins 14-3-3 and SHP-2[27,29], we also investigated the effect of *PIK3R1* loss on these interactions. However, there was no change in binding of Gab2 with these proteins following *PIK3R1* depletion as shown by immunoprecipitation and PLA (Supplementary Fig. 5d and Supplementary Fig. 5e).

***PIK3R1* loss activates AKT through multiple mechanisms**. Because *PIK3R1* loss could activate AKT in Gab2-deficient OVCAR5 cells (Fig. 4d), we explored the additional contributing mechanisms of AKT activation. AKT activity can be dependent on cellular PIP3 amount, which can be in turn determined by PI3K catalytic activity and PTEN level. Intriguingly, we observed increased PIP3 level in *PIK3R1*-depleted cells (Fig. 5a). Reduced PTEN protein level after *PIK3R1* loss was detected by RPPA (Fig. 2a) and subsequently by western blotting (Fig. 5b). Complementarily, cycloheximide chase assay showed that the stability of PTEN protein was reduced in *PIK3R1*-depleted cells (Fig. 5c). There was also significantly increase in the kinase activity of p110α (but not p110β) upon *PIK3R1* loss (Fig. 5d). We next examined the interaction between p110α and the adapter Gab2 or RTK. Indeed, the remaining p110α showed increased association with Gab2 or EGFR under *PIK3R1* depletion (Fig. 5e). AKT can be dephosphorylated by the protein phosphatases PHLPPL and PP2A. However, the protein levels of these AKT-inactivating phosphatases were not changed by *PIK3R1* knockdown (Supplementary Fig. 6).

**Activation of STAT3 and AKT by *PIK3R1* loss is independent**. Interconnection between STAT3 and PI3K signaling has been described[31,32]. We therefore examined the potential crosstalk between AKT and JAK2/STAT3 signaling downstream of *PIK3R1* loss. As shown in Fig. 6a, the AKT inhibitor MK-2206 attenuated AKT phosphorylation. Importantly, MK-2206 did not abrogate the induced phosphorylation of JAK2 and STAT3 upon *PIK3R1* loss. Reciprocally, to determine the effect of JAK2/STAT3 pathway on AKT activity, *PIK3R1* siRNA-transfected cells were treated with a JAK inhibitor AZD1480, which effectively inhibited STAT3 activity (Fig. 6b). *PIK3R1* siRNA still induced AKT phosphorylation in the presence of AZD1480 at statistically significant manner in three independent experiments (Fig. 6b). These observations obtained from pharmacological inhibition of AKT or STAT3 could be recapitulated by siRNA of the corresponding proteins (Supplementary Fig. 7). The results thus indicate that crosstalk between *PIK3R1* loss-induced activation of the STAT3 and AKT pathways is unlikely.

Because *PIK3R1* loss leads to PTEN destabilization and PTEN has been suggested to negatively regulate JAK2/STAT3 signaling[33–35], we further investigated whether PTEN reduction upon *PIK3R1* loss contributes to STAT3 activation. To this end, PTEN was overexpressed in *PIK3R1* siRNA-transfected cells. While the introduction of PTEN inhibited AKT phosphorylation as expected, it had no impact on JAK2 and STAT3 phosphorylation (Fig. 6c). In addition, immunoprecipitation showed that PTEN expression did not affect the formation of Gab2-mediated JAK2/STAT3 signalosome and PTEN itself was not in the complex (Fig. 6d). Therefore, PTEN is not involved in the *PIK3R1* loss-induced JAK2/STAT3 signaling axis.

***PIK3R1* loss drives sensitivity to STAT3 and AKT inhibitors**. Having established that *PIK3R1* loss activates STAT3 and AKT pathways in ovarian cancer cells, we reasoned that cells with *PIK3R1* loss would be sensitive to inhibitors of these two

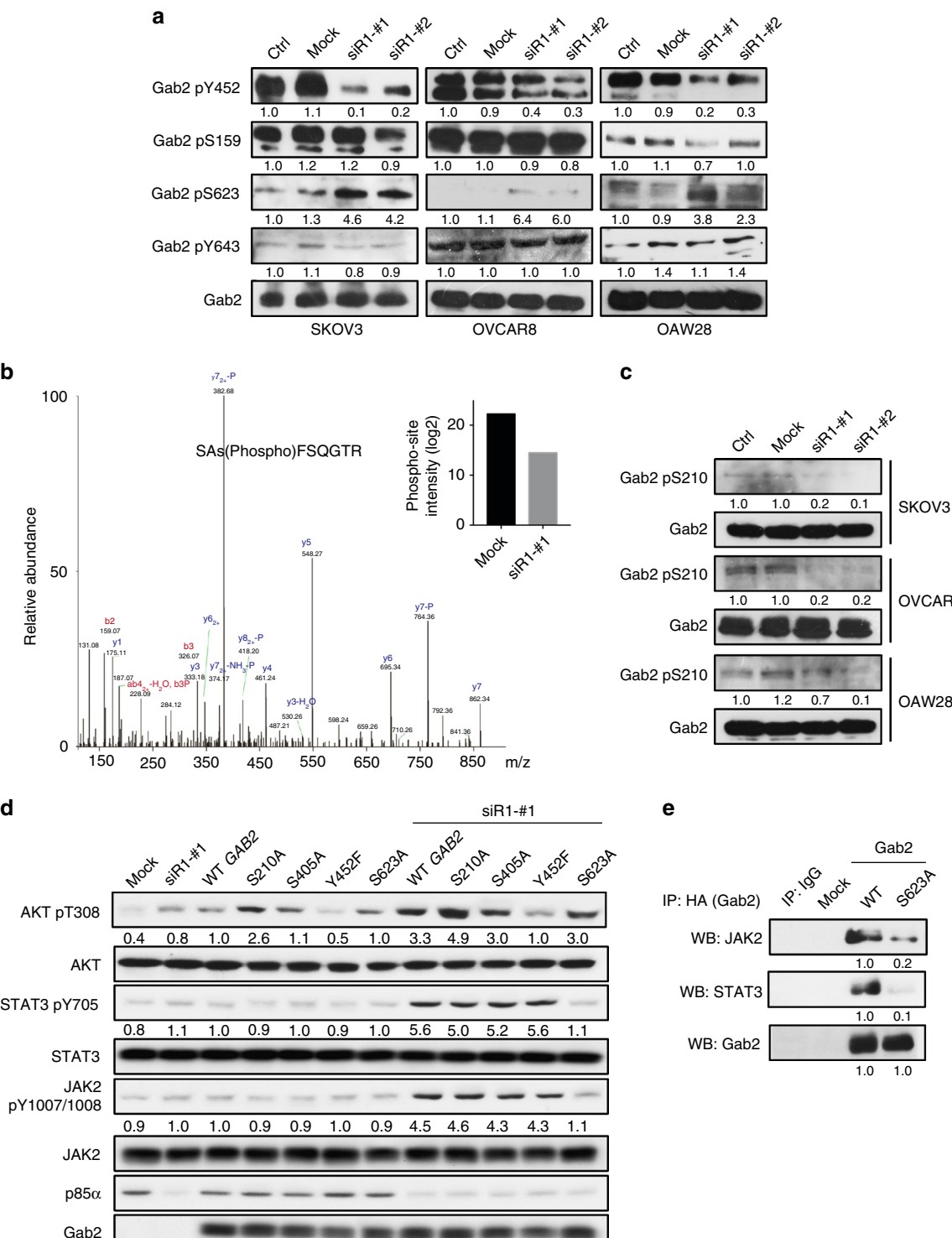

**Fig. 4** *PIK3R1* loss alters phosphorylation of Gab2 at S210 and S623, which in turn leads to signaling activation. **a** SKOV3, OVCAR8, and OAW28 cells were transfected with siRNA for 72 h prior to western blotting. **b** Protein lysates of HEK-293T transfected with *PIK3R1* siRNA or non-specific siRNA were subjected to immunoprecipitation with Gab2 antibody. The immunoprecipitates were subjected to SDS-PAGE separation for mass spectrometry-based quantitative phosphoproteomics. The MS/MS spectrum of phosphopeptide containing the S210 phosphorylated residue of Gab2 is shown. Inset, the log2 intensities of S210 phosphorylation in the samples. **c** SKOV3, OVCAR8, and OAW28 cells were transfected with siRNA for 72 h prior to western blotting. **d** OVCAR5 cells transfected with wild-type *GAB2* or *GAB2* mutants with or without *PIK3R1* siRNA for 72 h were harvested for western blotting. **e** OVCAR5 cells transfected with HA-tagged wild-type *GAB2* or *GAB2* S623A mutant were subjected to immunoprecipitation with HA antibody and western blotting. Cells transfected with empty vector but not HA-tagged Gab2 was used as control (labeled as mock). The numbers below the blots represent the mean values from densitometry readings of three independent experiments

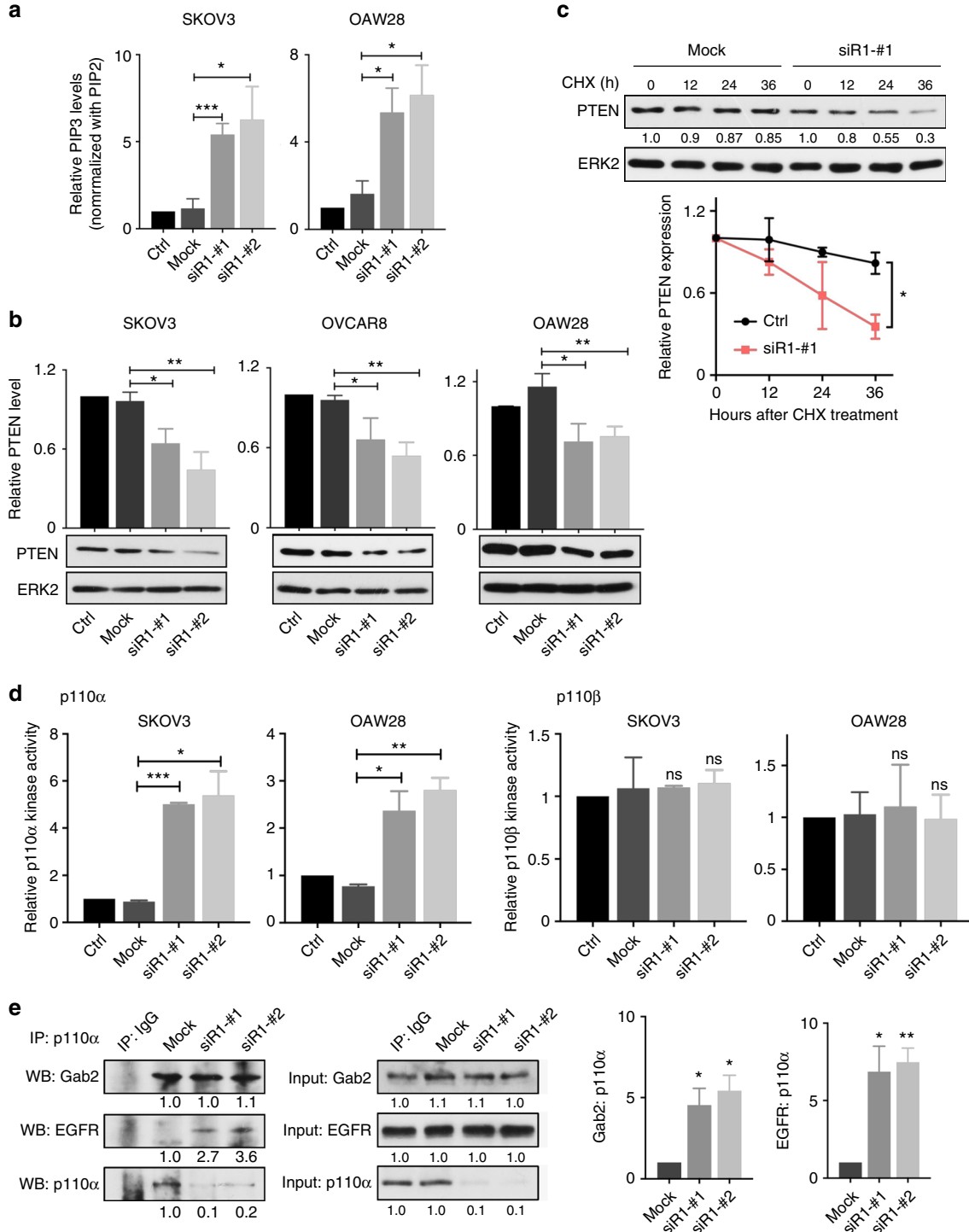

**Fig. 5** *PIK3R1* loss increases cellular PIP3 level by decreasing PTEN protein stability and enhancing p110α activity. **a** Lipids of transfected SKOV3 and OAW28 cells were collected and subjected to lipid detection. PIP3 level in each sample was normalized to that of PIP2. **b** Protein lysates were subjected to western blotting. Band intensities of blots from three independent experiments were quantified and normalized to that of mock. Mean values of relative PTEN levels with SD are shown. **c** siRNA-transfected SKOV3 cells were incubated with 50 μM Cycloheximide (CHX) for indicated hours before western blotting. Band intensities were normalized to 0 h time point and mean values with SD are shown. **d** p110α and p110β proteins were immunoprecipitated and subjected to PI3-kinase activity assay. Values were normalized to corresponding p110α and p110β protein levels. **e** p110α protein was immunoprecipitated in OAW28 cells transfected with *PIK3R1* siRNA prior to western blotting. The numbers below the blots represent the mean values from densitometry readings of three independent experiments. Error bars represent SD. *$p < 0.05$; **$p < 0.005$; ***$p < 0.001$; ns, no significant difference compared with mock using *t*-test

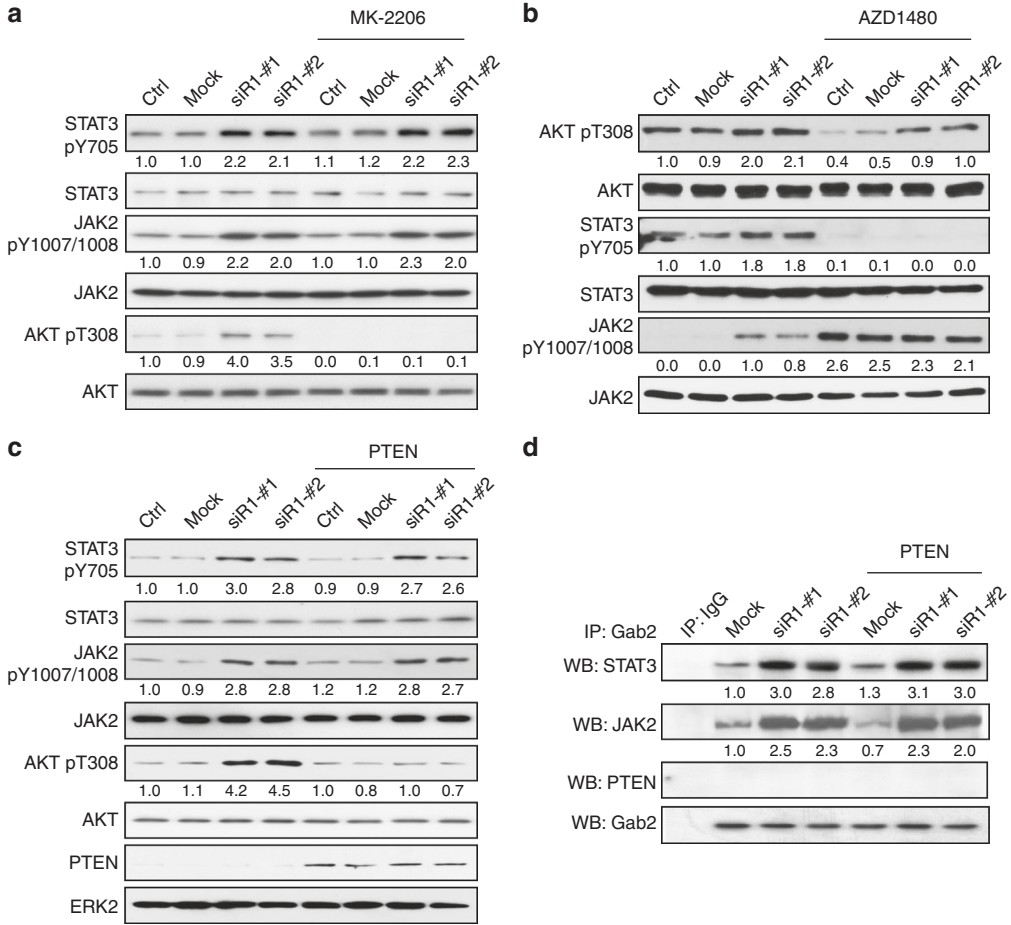

**Fig. 6** There is no crosstalk between *PIK3R1* loss-induced STAT3 and AKT signaling. **a** Transfected SKOV3 cells were treated with MK-2206 (2 μM) or DMSO control for another 24 h. Cell lysates were then subjected to western blotting. **b** Transfected SKOV3 cells were treated with AZD1480 (5 μM) or DMSO for 24 h before being harvested for western blotting. **c** SKOV3 cells co-transfected with *PIK3R1* siRNA and PTEN overexpression plasmid for 72 h were harvested for western blotting. **d** Protein lysates of SKOV3 co-transfected with *PIK3R1* siRNA and PTEN overexpression plasmid were immunoprecipitated with Gab2 antibody prior to western blotting. The numbers below the blots represent the mean values from densitometry readings of three independent experiments

pathways. Cellular responses to drug treatments in 3D culture may be more similar to what occurs in vivo and in patients compared to 2D culture. We therefore assessed the sensitivity of 3D spheroid ovarian cancer cells to the inhibition of AKT or JAK2/STAT3 signaling. Inhibitors of AKT (GDC-0068 and MK-2206), JAK2 (AZ960 and AZD1480), and STAT3 (C188-9 and Stattic) showed dose-dependent inhibitory effects on cell viability of SKOV3 and OVCAR8 spheroids (Fig. 7a and Supplementary Fig. 8a). Remarkably, spheroids transfected with *PIK3R1* siRNA were more sensitive to the inhibitors and demonstrated significantly lower IC50 values (Fig. 7a, b and Supplementary Fig. 8a, b). The sensitivity to AKT or STAT3 inhibitors of these cells was reversed in the presence of siRNA targeting the AKT isoforms or STAT3, respectively (Supplementary Fig. 8c, d). Concordant with the data that AKT and STAT3 pathways did not crosstalk, knockdown of AKT isoforms had no impact on sensitivity to STAT3 inhibitors. Likewise, *STAT3* siRNA did not affect responses to AKT inhibitors.

We next determined whether co-treatment with inhibitors against these pathways would enhance the therapeutic effect over that of each inhibitor alone. Cells were treated with serial concentrations of the AKT, JAK2 or STAT3 inhibitor alone or in combination with a fixed molar ratio. The combination of AKT with JAK2 or STAT3 inhibitors caused significantly more

reduction in cell viability than any single treatment (Fig. 7c and Supplementary Fig. 8e). We adapted the Chou-Talalay method for combination index (CI) to determine the effect of the drug combination (CI < 1, synergism; CI = 1, additive effect; CI > 1, antagonism). The CI values of combined AKT and JAK2 inhibitors or combined AKT and STAT3 inhibitors ranged from 0.395 to 0.823 (Fig. 7d and Supplementary Fig. 8f), indicating that the dual inhibition generates synergistic antitumor effect.

Responses to the inhibitors in vivo were evaluated in ovarian cancer xenografts injected i.p. with *PIK3R1*-shRNA stably expressing SKOV3 cells. Mice were treated intraperitoneally with vehicle control, AKT inhibitor (MK-2206), or STAT3 inhibitor (C188-9). Both inhibitors are in clinical trials. As shown in Fig. 8a, b, *PIK3R1*-shRNA tumors were significantly more sensitive to MK-2206 or C188-9 compared with tumors expressing vector control. The number and weight of peritoneal disseminated tumor nodules formed by SKOV3 cells expressing *PIK3R1*-shRNA, but not vector control, were markedly decreased after the inhibitor treatments. Western blotting confirmed that the *PIK3R1*-shRNA tumor nodules maintained decreased levels of p85α and increased levels of STAT3 and AKT phosphorylation (Supplementary Fig. 9a, b). Inhibition of STAT3 and AKT activities was observed in tumors harvested from mice treated with C188-9 and MK-2206, respectively (Supplementary

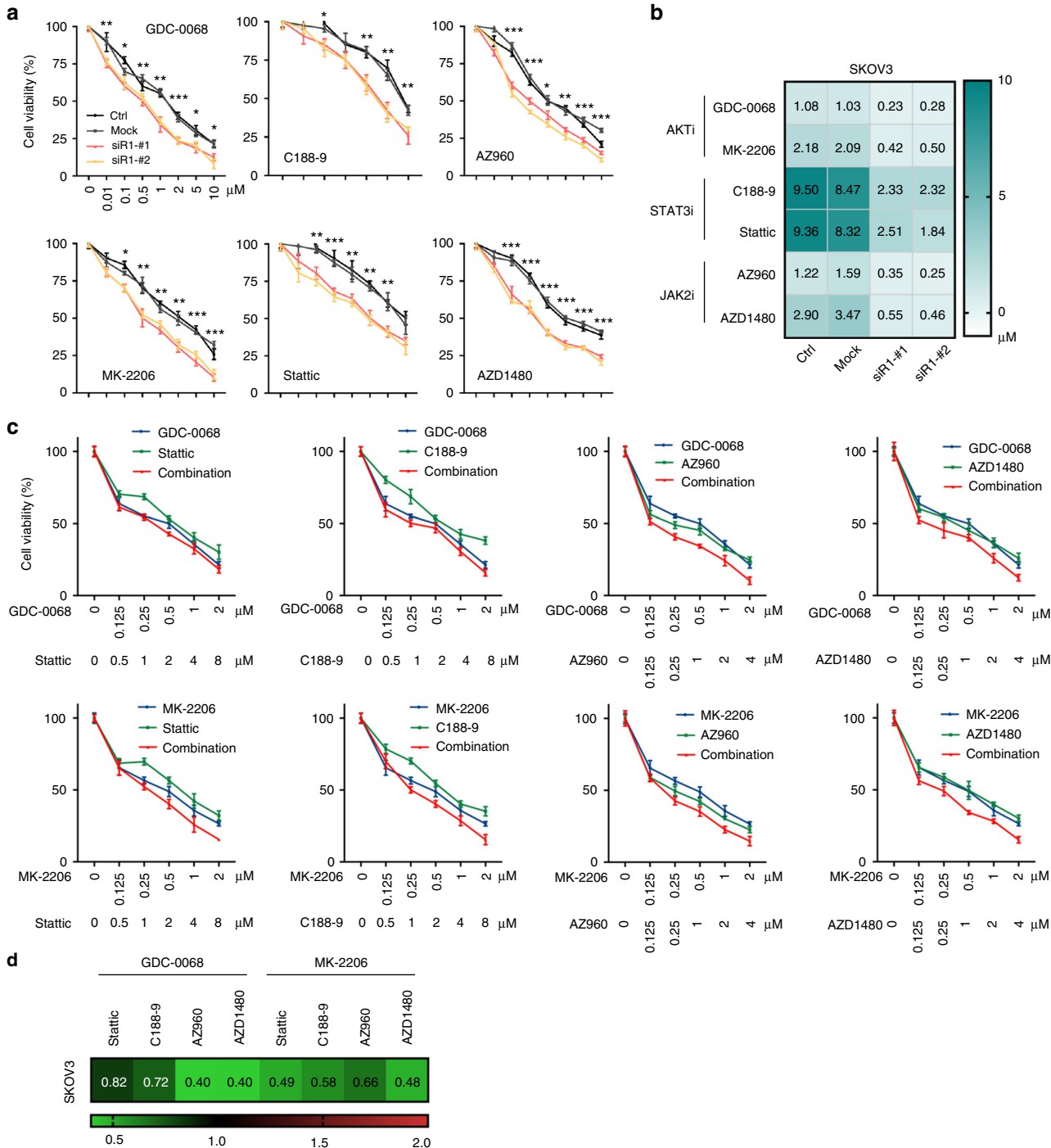

**Fig. 7** *PIK3R1* loss renders cells sensitive to JAK2, STAT3, and AKT inhibitors and co-targeting JAK2/STAT3 and AKT induces synergism. **a** SKOV3 cells transfected with *PIK3R1* siRNA were allowed to form 3D spheroids for 7 days prior to treatment with indicated inhibitors for 72 h. Dose–response curves of each inhibitor are shown. **b** IC50 values of each inhibitor were calculated by nonlinear regression analysis and are shown as heatmap. **c** SKOV3 3D spheroids were treated with each inhibitor alone or in combination as indicated, dose–response curves are shown. **d** Combination Index (CI) values were calculated with Chou-Talalay Method and CI values at the highest doses of the inhibitors used are visualized as heatmap. Error bars represent SD. *$p < 0.05$; **$p < 0.005$; ***$p < 0.001$ compared with mock using *t*-test

Fig. 9a, b), indicating the effectiveness and specificity of inhibitors in all treated mice. The more pronounced decrease in growth of *PIK3R1*-shRNA tumors implicated that these tumors were more dependent on the STAT3 and AKT signaling. We next sought to explore the response to the C188-9 and MK-2206 combination treatment in vivo. In line with the in vitro data, combination treatment with C188-9 and MK-2206 caused significantly

reduced tumor burden compared with C188-9 or MK-2206 treatment alone (Fig. 8c). Phosphorylation of STAT3 and AKT were effectively prohibited by the inhibitors alone or in combination (Supplementary Fig. 9c). Together, these results suggest that the dual inhibition of JAK2/STAT3 and AKT demonstrates striking antitumor effect in ovarian cancer cells with *PIK3R1* loss.

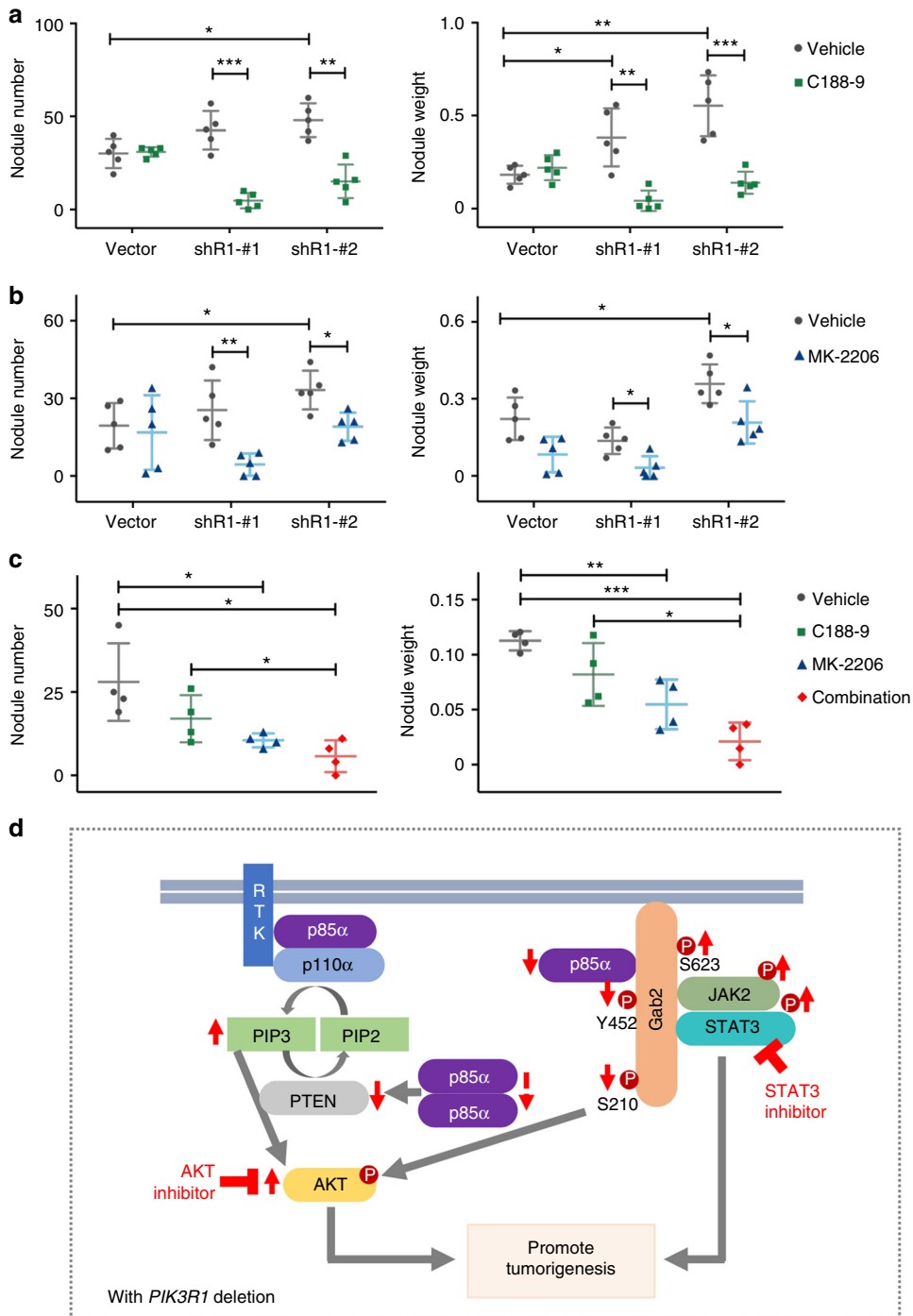

**Fig. 8** STAT3 and AKT inhibitors suppress *PIK3R1* loss-driven ovarian tumorigenesis. **a**, **b** Tumor-bearing mice ($n = 5$) were treated with vehicles, STAT3 inhibitor C188-9 (28 mg/kg) (**a**) or AKT inhibitor MK-2206 (75 mg/kg) (**b**) for 3 weeks. **c** Tumor-bearing mice ($n = 4$) were treated with vehicles, C188-9 (20 mg/kg) alone, MK-2206 (45 mg/kg) alone or in combination for 3 weeks. Bar graphs of tumor nodule number and weight are presented. Error bars represent SD. *$p < 0.05$; **$p < 0.005$; ***$p < 0.001$ using *t*-test. **d** Proposed working model: p85α in the PI3K heterodimeric complex stabilizes but inhibits p110, whereas p110-free 85α molecules form homodimers to stabilize PTEN. p85α is therefore a suppressor of AKT activation under normal condition. When p85α level reduces as a result of gene deletion in ovarian cancer, the abundance of p85α homodimer and thereby PTEN decrease. Increased PIP3 level is also caused by enhanced p110α activity, for example, through binding to receptor tyrosine kinase (RTK). Gab2 phosphorylation profile is changed including decreased S210 phosphorylation which involves in feedback inhibition of AKT and increased S623 phosphorylation facilitating JAK2/ STAT3 signalsome assembly. These events together lead to the activation of AKT and STAT3 signaling, which can potentially be exploited for therapy

## Discussion

Emerging evidence has indicated the functional consequences of *PIK3R1* loss in cancers. Activation of AKT has been demonstrated in the majority of cancer types studied except a prostate cancer cell line LNCaP in which AKT activity and cell proliferation is inhibited by p85α downregulation[11]. We also observed decreased AKT phosphorylation induced by *PIK3R1* siRNA in OVCAR3, which carries heterozygous *PIK3R1* mutation (M582 splice site at iSH2) that activates AKT. We therefore speculate that *PIK3R1* knockdown in these cells suppresses the

oncogenic p85α mutant and its downstream signaling. Intriguingly, LNCaP has a *PIK3R1* mutation R639* at the cSH2 domain which likely has similar effects to the mutation in OVCAR3. Truncation mutations at the iSH2 domain are thought to be oncogenic because they retain the ability to stabilize p110 while demonstrating reduced inhibition of p110. Whether this oncogenic mechanism can be extended to cSH2 domain truncation mutants warrants further examination. In *PIK3R1*-reduced renal cancer, activation of GSK3β/CTNNB1/Wnt pathway secondary to AKT signaling leads to cancer stem cell properties and enhanced expression of cadherins and EMT-related proteins[14]. Our findings herein demonstrated that reduced *PIK3R1* expression favors ovarian cancer tumorigenesis through AKT and AKT-independent activation of STAT3 signaling. However, we did not observe any changes in the levels of cadherins or EMT markers. Instead, the oncogenic phenotypes promoted by *PIK3R1* loss in ovarian cancer are likely mediated by matrix metalloproteinases and cell cycle proteins. This further suggests that *PIK3R1* loss promotes tumorigenesis through distinct mechanisms in different cancer types.

Activation of AKT is mediated through a cascade of upstream events. Signals from stimulated growth factor receptors are relayed to PI3K through multiple linker proteins including the docking protein Gab2[20,21]. Phosphorylated Gab2 associates with p85α and other signaling components. This recruitment event contributes to p110 activation, PIP3 production and subsequently AKT activation. p85α is known to stabilize p110 protein and p85-free p110 is prone to degradation[36]. Consistent with this notion and previous studies[37], we demonstrated decreased p110α protein level after *PIK3R1* abrogation. Our findings also revealed that *PIK3R1* loss results in decreased phosphorylation of p85α binding sites on Gab2 (Y452) that are critical for mediating PI3K and AKT activation. Given these decreases in p110α and phosphorylation of Gab2 Y452, the increased AKT activity upon *PIK3R1* loss may thus seem paradoxical. We propose that the activation of AKT is attributed to multiple mechanisms, including increase in PI3K activity, decreases in both PTEN expression and Gab2 S210 phosphorylation (Fig. 8d). This contribution of multiple mechanisms may reflect the complexity of regulation of the PI3K/AKT axis. We have reported that excess p85α molecules are p110-unbound and form homodimers that stabilize PTEN[6]. Aberrations such as *PIK3R1* mutation that disrupts homodimer formation or *PIK3R1* copy number loss that reduce p85α cellular amounts would lead to PTEN destabilization and increase in PIP3 levels. Cellular PIP3 level can also be increased by PI3K activity resultant from enhanced association with the RTK[38,39]. A previous study suggested that free p85α competes with p85-p110 heterodimer for binding to ErbB3 and this competition is relieved when p85α is reduced, promoting activation of PI3K[10]. Lastly, Gab2 S210 phosphorylation was shown to be initiated by AKT to mediate negative feedback, thereby inhibiting AKT signaling[29]. With AKT activated by *PIK3R1* loss, dephosphorylation of S210 may be a mechanism to abrogate the negative feedback. The identity of the phosphatases that dephosphorylate Gab2 S210 is completely unclear. It has been suggested that 14-3-3 binding to Gab2 S210 associates with the phosphorylation[29]. However, our data showed that the interactions between Gab2 and 14-3-3β and 14-3-3γ are unaltered by *PIK3R1* loss.

Aberrant activity of STAT3 plays a critical role in ovarian carcinogenesis and is associated with poor prognosis for ovarian cancer patients[40–42]. We characterized the mechanism by which STAT3 signaling is activated by *PIK3R1* loss. Although crosstalk between AKT and STAT3 has been suggested[31,32], *PIK3R1* loss-induced AKT signaling does not interact with STAT3 and vice versa. Rather, STAT3 signaling is mediated, at least in part, by

phosphorylated Gab2 at S623. Because of the ability to tether signaling molecules, Gab2 has been shown transmit signals to molecules such as STAT3[43]. So far two phosphorylation sites on Gab2 have been implicated in STAT signaling. STAT3 binds Gab2 at Y194 and this interaction activates STAT3 signaling upon stimuli from Stk in HEK-293 cells[25]. In a human IL-2-dependent T cell line Kit-225, phosphorylation of Gab2 S623 associates with STAT5 activation[27]. Interestingly, our data indicated that S623 is also critical for *PIK3R1* loss-induced STAT3 activation. It has been suggested that ERK phosphorylates S623 which results in reduced SHP-2 binding to Gab2 and increased STAT5 activation. First, our RPPA data showed no change in phosphorylation of SHP-2 Y542, which reflects SHP-2 phosphatase activity and signaling. Second, *PIK3R1* loss has no impact on association between Gab2 and SHP-2. These results suggested that STAT3 activation in the context of *PIK3R1* loss is distinct from STAT5 activation in Kit-225 cells. Concordantly, STAT5 is not activated in *PIK3R1*-depleted ovarian cancer cells. The regulatory mechanisms underlying changes in these Gab2 phosphorylation sites, particularly the kinase or phosphatase of these residues, have yet to be identified. Nonetheless, our data suggested that the alteration of S623 is independent of p110 activity as the p110 inhibitor has no effect on the induced STAT3 signaling. Further, it is noteworthy that *PIK3R1* loss activates AKT and STAT3 irrespective of *PIK3CA* (p110α) mutation status. One of the cell lines we used, SKOV3, carries the functional *PIK3CA* mutation (H1047R).

This study has direct translational significance. AKT inhibitors have entered clinical evaluation for multiple tumors types[44]. Preclinical studies of targeting STAT3 signaling by STAT3 or JAK2 inhibitors also showed potent antitumor activity in cancers and some of the inhibitors are in clinical trials[45–47]. The clinical effectiveness highlights the importance of biomarker-driven selection of patient subpopulations for clinical studies. Our findings showed that blockade of AKT or STAT3 may benefit ovarian cancer patients harboring *PIK3R1* copy number loss or reduced *PIK3R1* expression. Indeed, the effects are attenuated in parental cells with maintained *PIK3R1*. Whether *PIK3R1* expression in ovarian cancer is regulated by promoter hypermethylation or miRNA[48–50], which have been demonstrated to alter *PIK3R1* in other cancer types, remains to be explored. Further, combination of AKT and STAT3 inhibitors exerts synergistic antitumor effects in 3D spheroids in vitro and is more effective than either inhibitor alone in vivo. Drug synergism specifically between AKT and STAT3 inhibitor has not previously been reported. The closest example reported so far is a synergism between inhibition of pan class I PI3K and STAT3 in *KRAS* mutant-driven gastric cancer cells[51]. However, in contrast to the co-activation of AKT and STAT3 signaling in *PIK3R1* loss ovarian cancer cells, the molecular rationale underlying the synergism in the *KRAS* mutant gastric cancer cells is the activation of STAT3 upon inhibition of PI3K signaling. In summary, our comprehensive functional characterization of *PIK3R1* loss in ovarian cancer provides an explanation for the frequent *PIK3R1* deletion that occurs in the disease. We also provide a rationale for mechanism-based therapy involving dual inhibition of STAT3 and AKT signaling in ovarian cancer with *PIK3R1* loss or low p85α level. Future validation studies with supplementary cancer patient samples and models are warranted to fully explore the clinical application.

## Methods
**Cell culture and siRNA**. SKOV3, OVCAR8, and OVCAR5 were cultured in RPMI-1640 (Gibco, Carlsbad, CA) whereas OAW28 was cultured in DMEM (Gibco) supplemented with 5% fetal bovine serum (FBS; Gibco), 100 units/mL penicillin and 0.1 mg/mL streptomycin (Gibco) under a humidified atmosphere of

5% $CO_2$ at 37 °C. SKOV3 was obtained from American Type Culture Collection, OVCAR8 and OVCAR5 were from National Cancer Institute, and OAW28 was purchased from European Collection of Authenticated Cell Cultures. Cell lines were tested for mycoplasma and validated by short tandem repeat (STR) profiling.

All ON-TARGET plus siRNA and non-specific siRNA were purchased from Dharmacon (Lafayette, CO) and introduced into cells using Lipofectamine RNAiMAX (Invitrogen, Carlsbad, CA). Stable cells expressing *PIK3R1* shRNA were established by lentiviral transduction and puromycin selection. The sequences of siRNA and shRNA used are listed in Supplementary Table 1. *GAB2* mutants were generated by QuikChange Lightning Site-Directed Mutagenesis Kit (Agilent Technologies, Santa Clara, CA). All constructs were confirmed by sequencing.

**Cell proliferation assay.** Ovarian cancer cells were transfected with siRNA for 24 h before seeding into 96-well plate (1000 cells per well in triplicate). Cell proliferations were measured by BrdU cell proliferation assay kit (Cell Signaling, Danvers, MA) at indicated time-points. Briefly, cells were incubated with BrdU solution for 2 h and incorporated BrdU was detected by anti-BrdU mouse monoclonal antibody followed by HRP-linked secondary antibody. The absorbance readings at 450 nm of each well were obtained.

**Cell cycle assay.** Cells ($5 \times 10^5$) were serum starved overnight for synchronization prior to siRNA transfection for 48 h. Cells were harvested and washed with cold PBS twice, and then fixed in 1 mL of cold 70% ethanol for at least 2 h. After fixation, cells were resuspended and incubated with 100 μL RNase A solution (50 μg/mL; Sigma-Aldrich, St. Louis, MO) at 37 °C for 30 min. After incubation with 400 μL propidium iodide (PI; 20 μg/mL; Sigma-Aldrich), PI signals were obtained at 560 nm by flow cytometer Canto II analyzer (BD Biosciences, San Jose, CA). Cell cycle images were analyzed by FlowJo software (Tree Star Inc, Ashland, OR). Gating strategy is shown in Supplementary Fig. 10.

**Apoptosis assay.** Six hours after siRNA transfection, the cells ($5 \times 10^5$) were cultured in FBS-free media for another 48 h to induce apoptosis. Harvested cells were then resuspended with 100 μL Annexin-binding buffer, with 5 μL of FITC conjugated Annexin V (Invitrogen) and 1 μL PI solution (100 μg/mL) added to the solution at room temperature for 15 min. After incubation, Annexin V and PI signals were measured at 525 nm and 561 nm, respectively, by flow cytometer Canto II analyzer and data were analyzed by FlowJo.

**Migration and invasion assay.** siRNA-transfected cells ($3 \times 10^4$) suspended in serum-free medium were seeded into 1 mg/mL Matrigel-coated (invasion) or uncoated (migration) cell culture inserts (8-μm; Millipore, Billerica, MA). Medium with 10% FBS was added to the lower chambers. After 24 h, cells in the upper chambers were removed by cotton swab. Invaded or migrated cells attached to the other side of the inserts were fixed by methanol and stained by crystal violet. Five fields (100 ×) of each insert were randomly captured and cells numbers were counted.

**3D cell culture chemosensitivity assay.** Wells of 96-well plates were pre-coated with growth factor-reduced Matrigel (10 mg/mL; BD Biosciences). siRNA-transfected ovarian cancer cells were suspended in 2% Matrigel-containing media and seeded in the Matrigel-coated 96-well plates at 1000 cells/well density in triplicate. Cells were allowed to grow for 7 days before treatments with each indicated inhibitors alone at serial diluted concentrations or in combination at fixed molar ratio for 72 h in serum-reduced condition (0.5% FBS). The inhibitors were obtained from Selleckchem (Houston, TX). DMSO was used as control. Viable cells were detected by luminescent cell viability assay using CellTiter-Glo® 3D cell viability assay according to manufacturer's protocols (Promega Corporation, Madison, WI). IC50 values (50% inhibitory concentration) of single drug were determined through nonlinear regression analysis by GraphPad Prism 7 software (La Jolla, CA). Drug synergy was determined by the algorithm derived by Chou and Talalay[52,53] using the CalcuSyn software (Biosoft, Version 2.1). Combination index (CI) = 1: additive effect between the two agents, CI < 1: synergism and CI > 1: antagonism.

**In vivo tumorigenic and drug sensitivity assays.** All animal procedures were carried out with the approval of the Ethics Committee (Committee on the Use of Live Animals in Teaching and Research) of University of Hong Kong. SKOV3 cells ($5 \times 10^6$) that stably expressing *PIK3R1* shRNA or empty vector suspended in PBS were intraperitoneally injected into 6-week-old female nude mice (Charles River Lab, USA). Tumor-bearing mice were sacrificed after 5 weeks. Tumor nodules throughout the peritoneal cavity (including intestine, mesentery, liver, and spleen) were collected, counted, and weighted.

For the evaluation of antitumor effects by inhibitors, the animals were randomly divided into subgroups ($n = 5$ per group), 2 weeks after tumor cell inoculation and received drug suspension or vehicle through oral gavage 3 times/week for 3 weeks. C188-9 (28 mg/kg) was dissolved in water with 5% D-glucose and 5% DMSO, while MK-2206 (75 mg/kg) was dissolved in 30% Captisol (CyDex Pharmaceuticals, Lawrence, KS). The mice were euthanized when the experiments

reached the treatment endpoint and tumor burden were compared among subgroups. Proteins were harvested from the tumor nodules for examination of signaling changes by western blotting.

**Analysis of proteins.** Cells were lysed in RIPA buffer (150 mM NaCl, 0.1% SDS, 1% NP-40, 1% sodium deoxycholate) supplemented with protease and phosphatase inhibitors (Thermo Fisher Scientific, Waltham, MA). Total protein concentrations were determined using BCA kit (Thermo Fisher Scientific) prior to separation by 8% SDS-PAGE and transfer onto PVDF membranes (Amersham Hybond, GE Healthcare Life Science). After blocking by 5% non-fat milk, membranes were incubated with indicated primary antibodies overnight at 4 °C and corresponding HRP-conjugated secondary antibodies. Immunoreactive signal were detected using chemiluminescence detection kit (Bio-Rad, Hercules, CA). All antibodies used in this study are listed in Supplementary Table 2. Uncropped scans of immunoblots are provided in Supplementary Fig. 11–19.

For IP, cells were lysed in IP lysis buffer (50 mM Tris, 150 mM NaCl, 0.5% NP-40, 5 mM EDTA) supplemented with protease and phosphatase inhibitors. Protein lysate was then incubated with corresponding antibody overnight followed by incubation with protein A/G agarose beads (Santa Cruz Biotechnology, Dallas, TX). Unbound proteins were washed away with IP lysis buffer and immunoprecipitated proteins were eluted with 2X Laemmli sample buffer.

Cytosolic and nuclear fractions were prepared by Nuclear/Cytosol Fractionation Kit (Biovision, Milpitas, CA). Briefly, $2 \times 10^6$ cells were trypsinized and resuspended with Cytosol Extraction Buffer A. Cytosol Extraction Buffer B were added to the suspension followed by centrifugation to obtain cytosolic protein-containing supernatant whereas the pellet was further processed for nuclear proteins.

**Immunofluorescence.** SKOV3 cells ($3 \times 10^4$) were seeded onto sterilized cover slips and were transfected with *PIK3R1* siRNA. Twenty-four hours after transfection, the cells were harvested and treated with 4% paraformaldehyde for fixation, 0.1% Triton X-100 for permeabilization and 3% BSA for blocking unspecific binding. Then, the cells were incubated with primary antibody overnight and secondary antibody for 1 h in dark. Slides were mounted with coverslip in the mounting medium with DAPI. Images were captured under Carl Zeiss LSM 700 (Zeiss, Jena, Germany) with 461 nm (DAPI) and 594 nm (Texas Red). Representative images of each sample were selected from five randomly captured images.

**Proximity ligation assay.** SKOV3 cells ($4 \times 10^4$) were seeded onto sterilized cover slips and transfected with *PIK3R1* siRNA. Interaction signals detected by proximity ligation assay were performed according to the instruction of Duolink PLA fluorescence kit (Sigma-Aldrich). Briefly, cells were fixed by 4% paraformaldehyde, followed by 0.1% Triton X-100 permeabilization and blocking. Next, cells were incubated with a mixture of two primary antibodies overnight. As negative control, one of the antibodies would be replaced by IgG. Then, the cells were incubated with PLA probe (PLUS and MINUS) for 1 h. Probe ligation was performed with two circle forming DNA nucleotides by ligation-ligase solution. PLA signals were amplified using amplification-polymerase solution. Slides were mounted with coverslip in the mounting medium with DAPI. Fluorescence signals were observed at 461 nm (DAPI) and 594 nm (Texas Red) by Carl Zeiss LSM 700 (Zeiss). For each set of samples, at least three fields were captured and spots per nucleus were counted. Experiments were performed at least three times independently.

**Reverse phase protein array.** RPPA performed by our group has been described previously[5,54]. Briefly, after cell lysis as described above, protein concentration was adjusted to 1.5 μg/μl and denatured total proteins were serially diluted and printed onto nitrocellulose-coated slides. The slides were blocked, antibody-incubated and visualized by DAB colorimetric reaction. Slides were scanned and the densities of the spots were quantified by Array-Pro Analyzer. Relative protein levels for each sample were determined by "Supercurve Fitting" with a logistic regression model (http://bioinformatics.mdanderson.org/Software/supercurve). All data points were normalized for protein loading.

**Immunohistochemistry.** Archival formalin-fixed paraffin-embedded tissue blocks were retrieved from the Department of Pathology, Queen Mary Hospital, the University of Hong Kong, with the approval of Institutional Review Board of the University of Hong Kong/Hospital Authority Hong Kong West Cluster (HKU/HA HKW IRB). All cases were primary ovarian carcinoma of serous histotype and were obtained before the treatment with informed consent. Hematoxylin and eosin (H&E) staining of the specimens were reviewed by a pathologist (Annie NY Cheung) for ≥ 80% tumor content. The 5-μm sections were deparaffinized and rehydrated through graded alcohols. Antigen retrieval was performed using citrate buffer pH 6.0 (for p85α and p-AKT antibodies, Santa Cruz Biotechnology) or EDTA pH 8.0 (for p-STAT3 antibody, Cell Signaling). The sections were incubated in 3% $H_2O_2$ to reduce endogenous peroxidase activity and were blocked with goat serum. Incubation with primary antibody (1:20 for p85α and p-AKT and 1:50 for p-STAT3) was performed overnight at 4 °C followed by biotin-conjugated secondary antibody (Dako, Carpinteria, CA) for 1 h at room temperature. HRP was detected by DAB (Amresco, Solon, OH). Protein expression was represented by

histoscore score (0–6), which was calculated for each section by adding the score of percentage of positive cells (0 ≤ 5%, 1 = 6–25%, 2 = 26–50%, and 3 ≥ 51%) and the intensity score: 0 (negative), 1 (weak), 2 (moderate), and 3.

**Mass spectrometry-based phosphopeptide quantification**. HEK-293T cells transfected with *GAB2* expression plasmid with or without *PIK3R1* siRNA were lysed in IP lysis buffer. Ninety milligram total protein lysates were subjected to IP using Gab2 antibody (Santa Cruz Biotechnology) and the IP eluate was subjected to in-gel digestion and peptide extraction. Briefly, gel slices were subjected to reduction and alkylation by 8 mM DTT and 40 mM iodoacetamide, respectively. Protein digestion was performed by incubating with trypsin (1 ng/µl) overnight at 37 °C. Subsequent tryptic peptides were extracted from the gel with 100% ACN/3% TFA and 40% ACN/3%TFA. The peptide extracts derived from control or treatment group were desalted using C18 ZipTips for LC-MS/MS analysis with Dionex Ultimate3000 nanoRSLC system coupled to Thermo Fisher Orbitrap Fusion Tribid Lumos. Peptides were separated on commercial C18 column (75 µm i.d. × 50 cm length) with 1.9 µm particle size (Thermo Fisher).

Raw mass spectrometry data were processed using Maxquant version 1.6.0.1, with a false-discovery rate < 0.001 at the level of protein, peptides and modifications, using settings as below: oxidized methionine (M), acetylation (Protein N-term), and phospho (STY) were selected as dynamic modifications, and carbamidomethyl (C) as fixed modifications with minimum peptide length of seven amino acids was enabled. Proteins and peptides were identified against the Human UniProt FASTA database (July 2017) containing 28,250 entries. Quantification of peptides and proteins was performed by Maxquant. Bioinformatics analysis was performed with Perseus version 1.6.0.2 using label free quantitation (LFQ) protein intensities and ratios. For the analysis of phosphopeptides, 1% FDR, a minimum localization probability of 0.95 and a delta score of 40 were considered. Normalized phosphosite intensities (log2) from control and treatment group were used to calculate the site level expression.

**Luciferase assay**. A STAT3-luciferase reporter construct (4xM67 pTATA TK-Luc) containing four copies of STAT3-binding sites upstream of the minimal TK promoter was a gift from Jim Darnell (Addgene plasmid # 8688). Cells were transfected with *PIK3R1* siRNA for 24 h prior to co-transfection of 4xM67 pTATA TK-Luc with Renilla luciferase vector (pRL-TK) using Lipofectamine 3000 (Invitrogen). After another 24 h, cells were harvested for luciferase reporter assay using the dual-luciferase reporter assay system (Promega) following manufacturer's protocol. Firefly luciferase values were normalized to Renilla luciferase values.

**Real-time PCR**. Total RNA was extracted using TRIzol reagent (Life Technologies, Carlsbad, CA) and quantified on Nanodrop 2000 (Thermo Scientific). Five microgram of total RNA were reverse transcribed using SuperScript IV and oligo (dT) primers (Life Technologies). Real-time PCR was performed with Power SYBR Green PCR Master Mix according to manufacturer's instructions (Applied Biosystems, Foster City, CA) using a ABI Prism 7900 HT Sequence Detector (Applied Biosystems). The abundance of mRNA was determined using the $\Delta\Delta CT$ method (where *CT* is threshold cycle) with *GAPDH* as internal control. Sequences of the primers used are shown in Supplementary Table 3.

**PI3-kinase activity assay**. *PIK3R1*-siRNA-transfected SKOV3 and OAW28 cells were harvested and subjected to immunoprecipitation to obtain p110α or p110β proteins. Kinase activity assays were performed following the instruction of Echelon PI3-Kinase Activity Elisa: Pico kit (K-1000s, Echelon Biosciences, Salt Lake City, UT). Briefly, the kinase proteins were incubated with PI(4,5)P2 substrate at 37 °C before incubation with PI(3,4,5)P$_3$ detector. The samples were then transferred to Detection Plate followed by secondary detector. TMB solutions and H$_2$SO$_4$ stop solutions were subsequently added for color development. Absorbance readings were obtained at 450 nm. PIP3 levels were calculated using Sigmoidal dose–response nonlinear regression standard curves generated by GraphPad Prism. Relative PIP3 levels were normalized with the corresponding kinase protein levels. Experiments were performed in triplicates.

**PIP$_3$/P1P$_2$ quantification**. For lipid extraction, $2 \times 10^7$ SKOV3 or OAW28 cells were treated with *PIK3R1* siRNA and lipids were extracted by ice cold 0.5 M TCA. Neutral lipids were eliminated by MeOH:CHCl$_3$ (2:1) and acidic lipids were released by MeOH:CHCl$_3$:12 M HCl (80:40:1). The acidic lipids were then split into organic and aqueous phases by CHCl$_3$ and 0.1 M HCl. The organic phases (acidic lipids) were dried and reconstituted by PBS-T which were further subjected to lipid detection according to manufactory's instruction (K-2500s and K4500, Echelon Biosciences). Relative PIP$_3$ levels of samples were normalized by PIP$_2$ detected in parallel. Each value represents data of triplicates.

**Statistical analysis**. The experiments were repeated at least three times. Cell proliferation assay, STAT3 activity assay, RT-PCR assay, and drug sensitivity assay were performed in triplicate and all data were expressed as means ± SD. Significance was assessed by Student's *t*-test using GraphPad Prism. All *p*-values were two sided and *p* < 0.05 was considered significant.

**Reporting summary**. Further information on experimental design is available in the Nature Research Reporting Summary linked to this article.

## Data availability

The data that support the findings of this study are available from the corresponding author upon reasonable request.

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

## Acknowledgements

This study was supported by Hong Kong Research Grants Council (#27103616) and National Natural Science Foundation of China (#81703066) to L.W.C.; NCI (1P50 CA217685-01, 1U01 CA217842-01), Ovarian Cancer Research Foundation, a kind gift from the Adelson Medical Research Foundation to G.B.M.; Health and Medical Research Fund, Hong Kong Special Administrative Region (#03143006) to A.N.C.; National Natural Science Foundation of China (#81772777) and Shanghai Pujiang Program (17PJ1401400) to C.W. The authors would like to thank the RPPA Core Facility and the Characterized Cell Line Core Facility (funded by NCI #CA016672) of the MD Anderson Cancer Center (Houston, TX) and Dr. Takashi Saito (RIKEN Center for Integrative Medical Science, Japan), Dr. Gen-sheng Feng (University of California San Diego, USA), Dr. Susan Meakin (University of Western Ontario, Canada) for generously providing the wild-type *GAB2* plasmids. We also thank the Faculty Core Facility of the LKS Faculty of Medicine HKU for the help with flow cytometry and confocal microscopy.

## Author contributions

L.W.C. conceived and coordinated the project. X.L., V.C.M., Y.Z., C.W., Y.L., and L.W.C. performed experiments. X.L., V.C.M., G.B.M., and L.W.C analyzed the data. R.S. contributed to mass spectrometry analysis. E.S.W. and A.N.C. collected the tumor samples, validated tumor histology and prepared paraffin-embedded tissue sections. L.W.C. wrote the manuscript, with input from the other authors. All authors provided comments.

## Additional information

**Competing interests:** G.B.M. receives sponsored research support and is a member of the SAB of AstraZeneca. The other authors declare no competing interests.

