## [Peer Review File · Nature Communications]

Reviewers' comments:

Reviewer #1 (Remarks to the Author):

The PI3K/AKT/mTOR is a well-established signaling pathway involved in cell survival, growth and metabolism. Dysregulation of the pathway (mostly activation) contributes to the development and progression of various cancers types including ovarian cancer. In the current study, the authors report that loss of PIK3R1 (p85 α) in ovarian cancer cells results in activation of AKT and JAK2/STAT3 signaling- a process that is mediated by changes in the phosphorylation of the docking protein gab2. Moreover, PI3KR1 loss renders ovarian cancer cells vulnerable to inhibitors of AKT or JAK/STAT in vitro and in vivo. Finally, combination therapy of AKT and STAT3 increased anti-tumor effects as compared to single-agents treatments.

Overall, this is a well-written manuscript, which contains interesting and novel data regarding the role of PIK3R1 in the regulation of AKT as well as the potential involvement of Gab2 in that process. In addition, it nicely shows that JAK/STAT are activated following PIK3R1 knockdown, which is accompanied by significant reduction in cell proliferation and migration in vivo and in vitro. However, some controls are missing and in some cases the conclusions are not proven well enough. These and other comments are as follows:

- 1) The authors propose that induction of p-AKT level following siPIK3R1 is a consequence of reduction in PTEN level that is less stable. Although in the Reverse phase protein array the expression of PTEN is clearly reduced, it is much less evident using WB and need quantification and statistical analysis. The same is true for other lanes in figure 2B,c. Moreover, the stability of PTEN should be measured in other methods (i.e: cycloheximide). Finally, the actual activity of PI3K lacking the interacting p85 and the possible involvement of PP2A/PHLPP need to be sassed.
- 2) Gab2 phosphorylation at S210 inhibits AKT activation by a negative feedback loop. In the manuscript, there is no proof for dephosphorylation of S210 upon PIK3R1 loss, and it is recommended to test it. If such exists, the identity of the phosphatases should be discussed.
- 3) Based on previous publications, Gab1 phosphorylation causes interaction with STAT3. This point should be better discussed in this manuscript
- 4) The results in Figs 4C and 5 are not always convincing. The results need to be quantitated and the significance of the changes should be mentioned. In addition, it is recommended to confirm some of the interaction using other methods (e.g. PLA).
- 5) It is recommended to show the Nuclear translocation of STAT in the cell examined.
- 6) The model in Fig. 7 is too complex, and the sequence of events is not very clear recommend to simplify it.

Reviewer #2 (Remarks to the Author):

In this study, the authors attempt to show a genetic connection between PIK3R1 loss and different signalling cascades in ovarian cancer. They demonstrate that both STAT3 and Akt are signalling are perturb and this occurs through Gab2, though in different mechanistic ways. A series of functional studies with xenografts do largely affirm that these two pathways in the context of PIK3R1 loss are important for the formation of tumor nodules in vivo.

1. The data in the manuscript is clearly presented. However, I raise the concern about novelty since all of these pathways have been implicated in ovarian cancer in some way. Moreover, the connection of Gab2 and STAT3 or Akt is not new. The drug susceptibility could be a newer aspects of this work but ultimately the lack of a clear mechanism precludes the manuscript as a significant advance in our understanding of signalling in ovarian cancer.

2. Some of the in vitro assays show modest changes (e.g. Fig. 1A, 3D – proliferation increases are

less than 2 fold) and though I do agree that there are statistically significant differences in vivo (Fig 1f), these data show a large degree of variation. Other data from the TCGA analysis barely reach significant (Supp. Fig. 2C, STAT3 expression) which are key generalizability of their findings.

3. In line with my general feeling from Point #2 is the broad reduction of signal transduction due to PIK3R1 copy number loss. Is this specific just to STAT3 (Fig 2)? All of the presented data here found increases in signaling pathways that was rescued by KO of PIK3R1. Authors should show some signaling pathways that did not change or were found to be increased as controls for this assay. In addition, they should show that PIK3R1 KO would reduce expression of these respective pathways. For example, is ERK/MAPK or other STAT pathway altered? The same comments are true for studies in Fig. 3 - does GAB2 KO block other signaling pathways?

4. What are the histological subsets being considered in their analysis? The 3 cell lines used in the study are generally considered HGSC, but the p85 high expressing group looks histologically different than the p85 low expressing group. This is also a small dataset (n=28) which may explain why there is no significant association between p85 expression and STAT3/Akt. Thus, these data should be independently validated and carefully enumerated using statistically appropriate methods (e.g. scoring, stratification of subtypes etc). Given this point, I'm quite surprised the authors went on to do so much additional molecular work to block STAT3 signaling. Why not Akt? This seems especially relevant given that they claim STAT3 and Akt do not cross talk in the absence of PIK3R1. The pharmacological studies later in the paper seem to confirm this. My point here is that despite some reasonably nice data showing these pathways don't converge, the mechanisms of PIK3R1 is unresolved and is not resolved in my opinion with the in vivo inhibitor studies. They also don't show the specificity of these drugs in their in vivo experiments.

5. Is there a meaningful difference in survival of the treated animals? Are the relevant pathways affected in the tumors of these mice? Are the results similar if these PIK3R1 KO tumors are genetically defective for Akt or STAT3?

Reviewer #3 (Remarks to the Author):

Li et al. offer a multi-faceted study of PIK3R1 (p85alpha) loss in ovarian cancer cell lines and patient samples. This work is clinically significant because it offers mechanistic insight into rewiring of the signaling network caused/accompanied by copy number loss of PIK3R1, which is the most common deletion found in ovarian cancer. On a more fundamental level, there is interesting information here that should draw interest from researchers working on the regulation and PI3K/Akt and JAK/STAT signaling pathways. The manuscript is reasonably well written and organized. Overall, the study offers an in-depth characterization of PIK3R1 loss in ovarian cancer cells, and I expect its impact to be high.

My main concerns revolve around the state of PI3K signaling in relation to GAB2 and Akt phosphorylation status. Previous work has established that p110alpha expression is limited by the expression of p85alpha, because p110alpha monomers are readily degraded; this implies that p85alpha is in excess, and there is evidence that excess p85alpha drives formation of p85alpha homodimers. Accordingly, depletion of p85alpha reduces p110alpha level; it follows that p85-p110 heterodimers are far less abundant in the cell. Comments:

1. Phosphorylation of GAB2 Y452, a known p85alpha binding site, is reduced in p85alpha-depleted cells. A likely explanation is that pY452 is protected from dephosphorylation by p85alpha binding. Is this true? One could test this by combining p85 depletion with expression of the higher-affinity p85 SH2 domain and testing whether or not reduction of pY452 is reversed.

2. The authors note that excess p85alpha competes with p85-p110 heterodimers for p85 binding sites, such as GAB2 pY452. With p85alpha depleted, there is presumably less p85-p110 heterodimers in the cell, as reasoned above, but there may be even less excess p85 (as a percentage of the total p85). Therefore, it is not clear how the binding of p85-p110 heterodimers to GAB2 pY452 is affected by p85alpha depletion. A co-immunoprecipitation of GAB2 and p110alpha, with and without p85alpha depletion, would address this point. If p110alpha recruitment to the complex were increased, or decreased much less than the loss of pY452 or of total p110alpha, it would go a long way towards explaining why Akt phosphorylation is enhanced. I say that because, at least on a whole-cell level, the reduction in PTEN abundance is rather modest (Fig. 2B).

3. How does p85alpha depletion affect PIP3 abundance? It is implied that PIP3 increases, but for the reasons raised in point 2 above, I think this is highly questionable. With less overall binding of p85 to GAB2 (and the p85-p110 heterodimer question unsettled), and only modest change in PTEN abundance, my hypothesis is that PIP3 is less not more abundant in p85-depleted cells. If so, this would shift the focus to how GAB2 S210 dephosphorylation apparently enhances Akt T308 phosphorylation.

4. One could argue that, even if PTEN activity is not changed much, and recruitment of p110 to the GAB2 complex is reduced, p85alpha depletion might still reduce competition for other p85-p110 binding sites and allow PIP3 abundance to increase (or only modestly decrease). Countering such an argument is the evidence presented in Fig. 4C. In OVCAR5 cells that lack GAB2 expression, mutation of Y542 to alanine reduces Akt T308 phosphorylation relative to expression of WT; this is also the case in p85alpha-depleted cells, although in the representative blot it appears that Akt phosphorylation is higher when Y542A GAB2 expression is coupled with p85alpha depletion. There are two flaws with this experiment. One is that there is no control/mock transfection to compare these levels to the Gab2-deficient background. Does p85alpha depletion affect this basal level? Can GAB2 enhance Akt phosphorylation even with Y542 mutated? Second, the authors need to replace Y542A with Y542F to best mimic the unphosphorylated tyrosine and cause the least disruption of GAB2 structure.

Other comments:

5. For co-immunoprecipitation experiments (both in the main paper and supplemental), negative controls are lacking. A non-specific IgG control and anti-HA IP of lysates lacking HA-GAB2. This could be a relevant issue for 14-3-3 pulldown in particular, since 14-3-3 proteins have been identified in the CRAPome.

6. I did not find detailed information about the antibodies used in this study. This apparent omission needs to be corrected. In particular I was looking for clarity about the specificity of the anti-14-3-3 antibodies for the various 14-3-3 isoforms.

7. p. 6: 'phosphorylated by IL-2 in T lymphocytes' should be changed to, e.g., 'phosphorylated in IL-2-stimulated T-lymphocytes.'

Point-by-point responses to referees' comments

We sincerely thank the reviewers for the review, their interest in our study and all the very insightful comments. The responses are listed point-by-point. Page, line and figure refer to the revised manuscript. We have also highlighted the changes (fonts in blue) in the manuscript text file. We hope the reviewers and the editors find this manuscript to be much improved and suitable for publication in Nature Communications.

Reviewer #1 (Remarks to the Author and Authors' Responses):

The PI3K/AKT/mTOR is a well-established signaling pathway involved in cell survival, growth and metabolism. Dysregulation of the pathway (mostly activation) contributes to the development and progression of various cancers types including ovarian cancer. In the current study, the authors report that loss of PIK3R1 (p85 α) in ovarian cancer cells results in activation of AKT and JAK2/STAT3 signaling- a process that is mediated by changes in the phosphorylation of the docking protein gab2. Moreover, PI3KR1 loss renders ovarian cancer cells vulnerable to inhibitors of AKT or JAK/STAT in vitro and in vivo. Finally, combination therapy of AKT and STAT3 increased anti-tumor effects as compared to single-agents treatments.

Overall, this is a well-written manuscript, which contains interesting and novel data regarding the role of PIK3R1 in the regulation of AKT as well as the potential involvement of Gab2 in that process. In addition, it nicely shows that JAK/STAT are activated following PIK3R1 knockdown, which is accompanied by significant reduction in cell proliferation and migration in vivo and in vitro. However, some controls are missing and in some cases the conclusions are not proven well enough. These and other comments are as follows:

1) The authors propose that induction of p-AKT level following siPIK3R1 is a consequence of reduction in PTEN level that is less stable. Although in the Reverse phase protein array the expression of PTEN is clearly reduced, it is much less evident using WB and need quantification and statistical analysis. The same is true for other lanes in figure 2B,c. Moreover, the stability of PTEN should be measured in other methods (i.e: cycloheximide). Finally, the actual activity of PI3K lacking the interacting p85 and the possible involvement of PP2A/PHLPP need to be assessed.

Response: First, we have quantified the relative band intensities of PTEN protein and the densitometry values of 3 independent experiments were analyzed (now **Fig. 5b**). The decrease in PTEN protein levels upon *PIK3R1* knockdown was statistically significant across the 3 cell lines. *PIK3R1* siRNA also led to decrease in PTEN protein stability which was measured by cycloheximide chase experiments (**Fig. 5c**).

Second, the band intensities of all western blots including those in Fig. 2b and 2c were measured by densitometry. The increases in phosphorylated AKT, STAT3 and JAK2 by *PIK3R1* siRNA were statistically significant and this is now mentioned in the text (**p.4, lines 15-16**).

Third, we have performed PI3K activity assay which showed that the kinase activity of

p110 α (but not p110 β) in *PIK3R1*-depleted cells was increased (**Fig. 5d**), albeit the cellular level of p110 α protein was decreased. This increase in activity is likely mediated by enhanced binding to RTK (such as EGFR) upon *PIK3R1* loss (**Fig. 5e**). Further, we have measured cellular PIP3 level which was indeed increased (**Fig. 5a**).

Lastly, the protein levels of protein phosphatases that dephosphorylate AKT including three PP2A subunits (A, B, and C) and PHLPP1 were detected by western blotting. Their expression levels were not altered by *PIK3R1* siRNA (**Supplementary Fig. 6**). Together, these new data provide clear mechanistic insight on AKT activation upon *PIK3R1* depletion. Our proposed model is now discussed (**p.10, lines 37-46 to p.11, lines 1-5**).

2) Gab2 phosphorylation at S210 inhibits AKT activation by a negative feedback loop. In the manuscript, there is no proof for dephosphorylation of S210 upon *PIK3R1* loss, and it is recommended to test it. If such exists, the identity of the phosphatases should be discussed.

Response: In the original manuscript, dephosphorylation of Gab2 S210 upon *PIK3R1* loss was revealed by mass spectrometry but was not confirmed by western blot. It was because Gab2 pS210 antibody was not commercially available. To validate the decrease, we have obtained a custom-made antibody, which was used in a previous report. Concordant with the data of mass spectrometry, we observed reduced level of phosphorylated Gab2 at S210 (**Fig. 4c**). The phosphatases of this particular residue have never been reported and remain completely unknown. This has now been discussed (**p11, lines 5-8**).

3) Based on previous publications, Gab1 phosphorylation causes interaction with STAT3. This point should be better discussed in this manuscript.

Response: Thanks for the comment. We have evaluated the protein expression of total Gab1 and Gab1 pY307 (which was implicated in STAT3 signaling). Their expression levels were not changed upon *PIK3R1* loss (**Supplementary Fig. 4b**). Additionally, we have performed gene silencing of *GAB1*. Knockdown of *GAB1*, unlike *GAB2*, could not abolish *PIK3R1* loss-induced AKT and STAT3 (**Supplementary Fig. 4a**), indicating that Gab1 unlikely mediates the signaling activation (**p.6, lines 1-5**).

4) The results in Figs 4C and 5 are not always convincing. The results need to be quantitated and the significance of the changes should be mentioned. In addition, it is recommended to confirm some of the interaction using other methods (e.g. PLA).

Response: Densitometry analysis of the bands are shown in **Fig. 4d** (previous Fig. 4c) and **Fig. 6** (previous Fig. 5). The significance of changes are mentioned in the text (**p.7, lines 1 and 9; p.8, line 11**). On a related note, we have performed siRNA experiments against AKT isoforms or STAT3 to confirm the lack of crosstalk between the two pathways (**Supplementary Fig. 7**). While *AKT1/2/3* siRNA abolished *PIK3R1*-induced AKT activity,

the activation of STAT3 and JAK2 remained. Likewise, *STAT3* siRNA had no effect on AKT activation.

Besides immunoprecipitation, as suggested, we have utilized proximity ligation assay (PLA) to confirm the enhanced interaction between Gab2, STAT3 and JAK2 upon *PIK3R1* loss (**Fig. 3g**). A lack of change in interaction between Gab2 and 14-3-3 isoforms was also confirmed by PLA (**Supplementary Fig. 5e**). The results of immunoprecipitation and PLA were highly consistent.

5) It is recommended to show the Nuclear translocation of STAT in the cell examined.

Response: In addition to western blotting-based subcellular fractionation as in the original manuscript, we have now included confocal immunofluorescence data showing an enhanced level of nuclear STAT3 upon *PIK3R1* loss (**Fig. 2f**).

6) The model in Fig. 7 is too complex, and the sequence of events is not very clear recommend to simplify it.

Response: Thank you for the suggestion. We have simplified the working model (**now Fig. 8d**).

Reviewer #2 (Remarks to the Author and Authors' Responses):

In this study, the authors attempt to show a genetic connection between *PIK3R1* loss and different signalling cascades in ovarian cancer. They demonstrate that both STAT3 and Akt are signalling are perturb and this occurs through Gab2, though in different mechanistic ways. A series of functional studies with xenografts do largely affirm that these two pathways in the context of *PIK3R1* loss are important for the formation of tumor nodules in vivo.

1. The data in the manuscript is clearly presented. However, I raise the concern about novelty since all of these pathways have been implicated in ovarian cancer in some way. Moreover, the connection of Gab2 and STAT3 or Akt is not new. The drug susceptibility could be a newer aspects of this work but ultimately the lack of a clear mechanism precludes the manuscript as a significant advance in our understanding of signalling in ovarian cancer.

Response: There are several novelties of this manuscript. First, it is the functional consequences of *PIK3R1* loss in ovarian cancer, in particular the activation of STAT3 signaling which is not a known downstream of p85 α signaling. Also, although the role of Gab2 in mediating STAT3 and AKT signaling has been reported, the regulation of Gab2 activity (phosphorylation) by *PIK3R1* or upon *PIK3R1* loss was completely unknown.

Further, we have now elucidated in more detail the regulatory mechanisms of AKT activation upon *PIK3R1* loss (**Fig. 5**). We have provided more evidence that *PIK3R1* loss-induced AKT activation is through at least three mechanisms: (1) increased p110 α kinase activity, (2) decreased PTEN level and (3) decreased Gab2 S210 phosphorylation leading to reduced negative feedback on AKT activation. The proposed mechanistic model is now discussed (**p.10, lines 37-46 to p.11, lines 1-5**). All these important findings lay the ground of mechanism-based therapy for ovarian cancer with *PIK3R1* loss.

2. Some of the in vitro assays show modest changes (e.g. Fig. 1A, 3D – proliferation increases are less than 2 fold) and though I do agree that there are statistically significant differences in vivo (Fig 1f), these data show a large degree of variation. Other data from the TCGA analysis barely reach significant (Supp. Fig. 2C, STAT3 expression) which are key generalizability of their findings.

Response: We have repeated the in vitro proliferation assays and included a longer time point (7 days), in which a more significant increase could be observed (3-4 fold increase in *PIK3R1*-depleted cells compared with control cells; **Fig. 1a, 3d; Supplementary Fig. 4c-d**). We have also repeated the in vivo experiments. The sample size was increased to 10 for more stringent statistical test. Consistent with our data in the original manuscript, *PIK3R1*-depleted ovarian cancer cells were significantly more tumorigenic (**Fig. 1e**).

3. In line with my general feeling from Point #2 is the broad reduction of signal transduction due to *PIK3R1* copy number loss. Is this specific just to STAT3 (Fig 2)? All of the presented data here found increases in signaling pathways that was rescued by KO of *PIK3R1*. Authors should show some signaling pathways that did not change or were found to be increased as controls for this assay. In addition, they should show that *PIK3R1* KO would reduce expression of these respective pathways. For example, is ERK/MAPK or other STAT pathway altered? The same comments are true for studies in Fig. 3 - does GAB2 KO block other signaling pathways?

Response: Our initial discovery of the increased STAT3 and AKT signaling upon *PIK3R1* loss stemmed from the protein array, in which major signaling pathways including MAPK were assayed. Indeed, only STAT3 and AKT were found to be activated and the other pathways were not changed (the data of these unchanged pathways are now included in **Supplementary Fig. 2a**). The activation status of the 3 MAPKs (ERK1/2, p38MAPK and JNK) were further confirmed by western blotting (**Supplementary Fig. 2c**). In addition, we now also demonstrated that STAT5 signaling (another STAT pathway implicated in cancers) was not changed by *PIK3R1* siRNA (**Supplementary Fig. 2c**). These results are mentioned in the text (**p.4, lines 18-20**). We therefore focused on STAT3 and AKT as well as the contribution of Gab2 in these two signaling pathways.

4. What are the histological subsets being considered in their analysis? The 3 cell lines used in the study are generally considered HGSC, but the p85 high expressing group

looks histologically different than the p85 low expressing group. This is also a small dataset (n=28) which may explain why there is no significant association between p85 expression and STAT3/Akt. Thus, these data should be independently validated and carefully enumerated using statistically appropriate methods (e.g. scoring, stratification of subtypes etc). Given this point, I'm quite surprised the authors went on to do so much additional molecular work to block STAT3 signaling. Why not Akt? This seems especially relevant given that they claim STAT3 and Akt do not cross talk in the absence of PIK3R1. The pharmacological studies later in the paper seem to confirm this. My point here is that despite some reasonably nice data showing these pathways don't converge, the mechanisms of PIK3R1 is unresolved and is not resolved in my opinion with the in vivo inhibitor studies. They also don't show the specificity of these drugs in their in vivo experiments.

Response: Thank you for the suggestions on the immunohistochemical analysis of our in-house ovarian cancer samples. Accordingly, we have now included ONLY the serous subtype (which is the histological subtype focused in this study) in our analysis. We have also increased the sample size to 43. The new analysis consistently showed negative correlation between the expression levels of p85 α and pAKT or pSTAT3, but at a more statistically significant manner (**p.4, lines 44-46 to p.5, lines 1-5**). The association between p85 α and p-STAT3 or p-AKT was $r=-0.53$ ($p=0.0002$) or $r=-0.4$ ($p=0.007$). The stained slides were reviewed by pathologist (A.N.C, a co-author). Representative images are shown in **Fig. 2d**.

As mentioned above, we have performed additional experiments that revealed the activation mechanism of AKT by *PIK3R1* loss (**Fig. 5**). Apart from showing that pharmacological inhibition of AKT or STAT3 did not affect the activation of STAT3 or AKT respectively as in the original manuscript (now **Fig. 6**), we have also performed new experiments with siRNA of AKT or STAT3, which again failed to block the activation of the other pathway (**Supplementary Fig. 7; p.8. lines 11-13**). These new data have strengthened our original data that the *PIK3R1* loss-induced AKT and STAT3 pathways do not crosstalk.

Regarding the specificity of the STAT3 and AKT inhibitors used in our in vivo experiments, we have shown in the original manuscript (previously Supplementary Fig. 7; now **Supplementary Fig. 9; p.9, lines 19-22, 27-29**) that the STAT3 inhibitor blocked STAT3 but had no effect on AKT activity, whereas AKT inhibitor was effective on blocking AKT and not STAT3. We have also added new drug response data that can further demonstrate the specificity of the inhibitors (**Supplementary Fig. 8c-d**; please see below comment #5).

5. Is there a meaningful difference in survival of the treated animals? Are the relevant pathways affected in the tumors of these mice? Are the results similar if these PIK3R1 KO tumors are genetically defective for Akt or STAT3?

Response: Two to three mice in the control groups appeared very unhealthy with weight loss at the scheduled end-point of the experiments. All the animals (including the treated ones for comparisons between groups) were sacrificed for humane reasons and survival

analysis was therefore not carried out.

Supplementary Fig. 7 in our original manuscript (now **Supplementary Fig. 9; p.9, lines 19-22, 27-29**) showed that STAT3 signaling was effectively inhibited by the STAT3 inhibitor which had no effect on AKT signaling. Likewise, the AKT inhibitor only blocked AKT signaling but not STAT3. Tumors of mice treated with both inhibitors had concurrent decrease in STAT3 and AKT signaling.

To further demonstrate the independency of the two signaling pathways and the specificity of the inhibitors, we have performed drug response assays using cells with siRNA targeting STAT3 or AKT isoforms, in the presence or absence of *PIK3R1* siRNA (**Supplementary Fig. 8c-d; p.8, lines 39-43**). Cells with STAT3 knockdown became less responsive to STAT3 inhibitors, whereas the sensitivity of these cells to AKT inhibitors was retained. AKT knockdown decreased the sensitivity of the cells to AKT inhibitors but not to STAT3 inhibitors.

Reviewer #3 (Remarks to the Author and Authors' Responses):

Li et al. offer a multi-faceted study of PIK3R1 (p85alpha) loss in ovarian cancer cell lines and patient samples. This work is clinically significant because it offers mechanistic insight into rewiring of the signaling network caused/accompanied by copy number loss of PIK3R1, which is the most common deletion found in ovarian cancer. On a more fundamental level, there is interesting information here that should draw interest from researchers working on the regulation and PI3K/Akt and JAK/STAT signaling pathways. The manuscript is reasonably well written and organized. Overall, the study offers an in-depth characterization of PIK3R1 loss in ovarian cancer cells, and I expect its impact to be high.

My main concerns revolve around the state of PI3K signaling in relation to GAB2 and Akt phosphorylation status. Previous work has established that p110alpha expression is limited by the expression of p85alpha, because p110alpha monomers are readily degraded; this implies that p85alpha is in excess, and there is evidence that excess p85alpha drives formation of p85alpha homodimers. Accordingly, depletion of p85alpha reduces p110alpha level; it follows that p85-p110 heterodimers are far less abundant in the cell. Comments:

1. Phosphorylation of GAB2 Y452, a known p85alpha binding site, is reduced in p85alpha-depleted cells. A likely explanation is that pY452 is protected from dephosphorylation by p85alpha binding. Is this true? One could test this by combining p85 depletion with expression of the higher-affinity p85 SH2 domain and testing whether or not reduction of pY452 is reversed.

Response: Thanks for the suggestion. We have examined this possibility which turned out to be very likely. Expression of full-length wild-type p85 α or p85 α SH2 domains (which is resistant to siRNA targeting the 3'UTR of the endogenous *PIK3R1*) in *PIK3R1*

knockdown cells could reversed the reduced phosphorylation of Gab2 Y452 (**Supplementary Fig. 5a**), implicating that the decreased phosphorylation could be an effect of reduced association with p85 α .

2. The authors note that excess p85 α competes with p85-p110 heterodimers for p85 binding sites, such as GAB2 pY452. With p85 α depleted, there is presumably less p85-p110 heterodimers in the cell, as reasoned above, but there may be even less excess p85 (as a percentage of the total p85). Therefore, it is not clear how the binding of p85-p110 heterodimers to GAB2 pY452 is affected by p85 α depletion. A co-immunoprecipitation of GAB2 and p110 α , with and without p85 α depletion, would address this point. If p110 α recruitment to the complex were increased, or decreased much less than the loss of pY452 or of total p110 α , it would go a long way towards explaining why Akt phosphorylation is enhanced. I say that because, at least on a whole-cell level, the reduction in PTEN abundance is rather modest (Fig. 2B).

Response: We have performed the suggested immunoprecipitation experiments. The binding between p110 α and Gab2, normalized to the input levels of p110 α , was increased in *PIK3R1*-depleted cells (**Fig. 5e**). We also found increased binding between p110 α and EGFR (**Fig. 5e**). The enhanced binding implicates that the activity of p110 α molecules was increased upon *PIK3R1* loss. This is supported by our new data obtained from the p110 α kinase assay in which p110 α showed increased kinase activity upon *PIK3R1* loss (**Fig. 5d**).

3. How does p85 α depletion affect PIP3 abundance? It is implied that PIP3 increases, but for the reasons raised in point 2 above, I think this is highly questionable. With less overall binding of p85 to GAB2 (and the p85-p110 heterodimer question unsettled), and only modest change in PTEN abundance, my hypothesis is that PIP3 is less not more abundant in p85-depleted cells. If so, this would shift the focus to how GAB2 S210 dephosphorylation apparently enhances Akt T308 phosphorylation.

Response: We have measured the levels of PIP3 (normalized to that of PI(4,5)P2 which is the control for total phosphoinositides). Cells with depleted *PIK3R1* had higher PIP3 levels (**Fig. 5a**)

4. One could argue that, even if PTEN activity is not changed much, and recruitment of p110 to the GAB2 complex is reduced, p85 α depletion might still reduce competition for other p85-p110 binding sites and allow PIP3 abundance to increase (or only modestly decrease). Countering such an argument is the evidence presented in Fig. 4C. In OVCAR5 cells that lack GAB2 expression, mutation of Y542 to alanine reduces Akt T308 phosphorylation relative to expression of WT; this is also the case in p85 α -depleted cells, although in the representative blot it appears that Akt phosphorylation is higher when Y542A GAB2 expression is coupled with p85 α depletion. There are two flaws with this experiment. One is that there is no control/mock transfection to compare these

levels to the Gab2-deficient background. Does p85alpha depletion affect this basal level? Can GAB2 enhance Akt phosphorylation even with Y542 mutated? Second, the authors need to replace Y542A with Y542F to best mimic the unphosphorylated tyrosine and cause the least disruption of GAB2 structure.

Response: To evaluate the contribution of Gab2 in *PIK3R1* loss-induced AKT activation, we have performed again the Gab2 mutant experiments in OVCAR5 (which lacks endogenous Gab2 protein expression). As suggested, parental and mock-transfected cells were included as controls; the Gab2 mutant Y452A was replaced by Y452F (**Fig. 4d**).

Several observations could be made: first, *PIK3R1* siRNA could increase pAKT in the control cells without Gab2 overexpression, suggesting that AKT activation by *PIK3R1* siRNA is less likely totally dependent on Gab2. In this regard, we have new data showing Gab2 siRNA had no effect on this activation in parental OVCAR5 (**Supplementary Fig. 5c**), confirming that Gab2 did not mediate this increase. PI3K can bind RTK without adaptor proteins and Gab2-independent AKT activation has been previously reported. In contrast, STAT3 signaling was not significantly activated by *PIK3R1* depletion in the parental OVCAR5 without Gab2 expression.

Second, consistent with the notion that *PIK3R1* loss leads to AKT activation, pAKT levels were higher in all cells with *PIK3R1* siRNA than the counterpart without the siRNA.

Third, the activation of AKT was inhibited in cells expressing Y452F, in the presence or absence of *PIK3R1* siRNA, indicating the critical importance of Y452 for Gab2 to mediate AKT activation.

Fourth, pAKT was higher in cells with Gab2 S210A and was further increased by *PIK3R1* knockdown. The increase of pAKT by *PIK3R1* siRNA even phosphorylation of S210 was prohibited implicates that relieving the negative feedback inhibition is not the sole mechanism of *PIK3R1* loss-induced AKT activation.

The data collectively suggest that the activation of AKT by *PIK3R1* loss is mediated by multiple mechanisms (**p.10, lines 37-46 to p.11, lines 1-5**).

Other comments:

5. For co-immunoprecipitation experiments (both in the main paper and supplemental), negative controls are lacking. A non-specific IgG control and anti-HA IP of lysates lacking HA-GAB2. This could be a relevant issue for 14-3-3 pulldown in particular, since 14-3-3 proteins have been identified in the CRAPome.

Response: We have repeated all the co-immunoprecipitation experiments with the suggested negative controls (**Fig. 3f, 4e, 5e, 6d; Supplementary Fig. 5d**). In addition, we have examined the interaction between Gab2 and 14-3-3 β and 14-3-3 γ using proximity ligation assay (PLA) assays (**Supplementary Fig. 5e**). The data obtained were consistent with that of the immunoprecipitation experiments.

6. I did not find detailed information about the antibodies used in this study. This apparent omission needs to be corrected. In particular I was looking for clarity about the specificity

of the anti-14-3-3 antibodies for the various 14-3-3 isoforms.

Response: We have now provided the information of all antibodies used in this study (**Supplementary Table 2**). The two anti-14-3-3 antibodies we included are said to be specific to the corresponding isoforms.

14-3-3 β (sc-25276): specific for an epitope mapping between amino acids 225-244 at the C-terminus of 14-3-3 β of human origin.

14-3-3 γ (sc-398423): specific for an epitope mapping between amino acids 128-149 within an internal region of 14-3-3 γ of human origin.

In addition, we have provided sequences of all siRNA, shRNA and primers used in this study (**Supplementary Tables 1, 3**).

7. p. 6: 'phosphorylated by IL-2 in T lymphocytes' should be changed to, e.g., 'phosphorylated in IL-2-stimulated T-lymphocytes.'

Response: Thank you. This has been revised (**p.6, line 42**).

REVIEWERS' COMMENTS:

Reviewer #1 (Remarks to the Author):

The authors addressed all my concerns. I find the paper ready for publication

Reviewer #2 (Remarks to the Author):

The authors have done a very nice job of revising the manuscript and have provided reasonable explanations for the novelty of their findings.

The remaining issue I have is the independent validation of the human ovarian tumor specimen in Figure 2d (IHC staining). While I appreciate that they have increased the sample size, focused on HGSC, and had a pathologist score it, these results need to be validated on an independent dataset. Sampling biases, differences in treatment of patients (based on jurisdiction), QC of sample collection, etc may have unintended impact on the tissue. What I don't want is the paper to lose sight of the fact that these results may have consequences on how we approach treating HGSC. This is the only human data presented in the manuscript and to this point, I think it needs to be strengthened.

Reviewer #3 (Remarks to the Author):

I am satisfied by the authors' thorough response to previous critiques. No concerns here.

Responses to referees' comments

Reviewer #1 (Remarks to the Author and Authors' Responses):

The authors addressed all my concerns. I find the paper ready for publication.

Response: We thank the reviewer for the time to review our manuscript and the support for publication.

Reviewer #2 (Remarks to the Author and Authors' Responses):

The authors have done a very nice job of revising the manuscript and have provided reasonable explanations for the novelty of their findings.

The remaining issue I have is the independent validation of the human ovarian tumor specimen in Figure 2d (IHC staining). While I appreciate that they have increased the sample size, focused on HGSC, and had a pathologist score it, these results need to be validated on an independent dataset. Sampling biases, differences in treatment of patients (based on jurisdiction), QC of sample collection, etc may have unintended impact on the tissue. What I don't want is the paper to lose sight of the fact that these results may have consequences on how we approach treating HGSC. This is the only human data presented in the manuscript and to this point, I think it needs to be strengthened.

Response: We thank the reviewer for the time to review our manuscript again and for acknowledging our efforts in improving the manuscript. The suggestion of validating the signaling changes in an independent dataset is well-taken. While time and effort are needed for the validation, this will not be included in this manuscript. We have added a statement at the end of the Discussion: "Future validation studies with supplementary cancer patient samples and models are warranted to fully explore the clinical application".

Reviewer #3 (Remarks to the Author and Authors' Responses):

I am satisfied by the authors' thorough response to previous critiques. No concerns here.

Response: Thank you. We are very glad to hear that the reviewer is satisfied with our revisions and has no further concerns.